# GAME-THEORETIC ROBUST RL HANDLES TEMPORALLY-COUPLED PERTURBATIONS

**Yongyuan Liang**[†][*]   **Yanchao Sun**[†]   **Ruijie Zheng**[†]   **Xiangyu Liu**[†]
**Benjamin Eysenbach**[§]   **Tuomas Sandholm**[‡]   **Furong Huang**[†]   **Stephen McAleer**[‡]
[†] University of Maryland, College Park   [‡] Carnegie Mellon University   [§] Princeton University

## ABSTRACT

Deploying reinforcement learning (RL) systems requires robustness to uncertainty and model misspecification, yet prior robust RL methods typically only study noise introduced independently across time. However, practical sources of uncertainty are usually coupled across time. We formally introduce temporally-coupled perturbations, presenting a novel challenge for existing robust RL methods. To tackle this challenge, we propose GRAD, a novel game-theoretic approach that treats the temporally-coupled robust RL problem as a partially-observable two-player zero-sum game. By finding an approximate equilibrium within this game, GRAD optimizes for general robustness against temporally-coupled perturbations. Experiments on continuous control tasks demonstrate that, compared with prior methods, our approach achieves a higher degree of robustness to various types of attacks on different attack domains, both in settings with temporally-coupled perturbations and decoupled perturbations.

## 1 INTRODUCTION

In recent years, reinforcement learning (RL) has demonstrated success in tackling complex decision-making problems in various domains. However, the vulnerability of deep RL algorithms to test-time changes in the environment or adversarial attacks has raised concerns for real-world applications. Developing robust RL algorithms that can defend against these adversarial attacks is crucial for the safety, reliability and effectiveness of RL-based systems.

In most existing research on robust RL (Huang et al., 2017; Liang et al., 2022; Sun et al., 2022; Tessler et al., 2019; Zhang et al., 2020), the adversary is able to perturb the observation or action every timestep under a static constraint. Specifically, the adversary's perturbations are constrained within a predefined space, such as an $L_p$ norm, which remains unchanged from one timestep to the next. This *standard* assumption in the robust RL literature can be referred to as a *non-temporally-coupled* assumption. However, this static constraint can lead to unrealistic perturbations: for example, the attacker may be able to blow the wind hard southeast at time $t$ but northwest at time $t + 1$. Providing robustness against such an perturbations may result in an overly conservative policy.

However, the set of perturbations faced in the real world are typically *temporally-coupled*: if the wind blows in one direction at one time step, it will likely blow in a similar directly at the next step. In this paper, we will treat the robust RL problem as a partially-observable two-player game and use tools from game theory to acquire robust policies, both for the non-temporally-coupled and the temporally-coupled settings.

In this paper, we propose a novel approach: Game-theoretic Response approach for Adversarial Defense ( GRAD) that leverages Policy Space Response Oracles (PSRO) (Lanctot et al., 2017) for robust training. GRAD is more general than prior adversarial defenses in the sense that it does not target certain adversarial scenarios and converges to the approximate equilibrium training with an adversary policy set. While prior methods often assume the worst case and aim to improve against them, they lack adaptability to specific attacks such as these adversaries under temporally-coupled constraints. We formulate the interaction between the agent and the temporally-coupled adversary as a two-player zero-sum game and employ PSRO to ensure the agent's best response against the learned

---

[*]Corresponding author. `cheryunl@umd.edu`

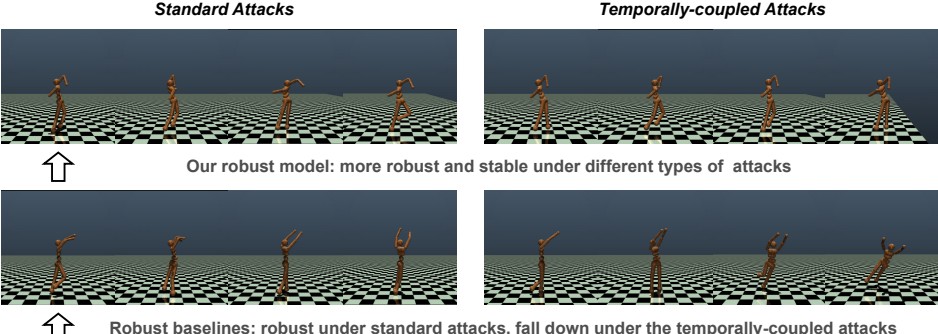

**Figure 1:** The robust `GRAD` agents *(top)* and the state-of-the-art robust WocaR-RL (Liang et al., 2022) *(bottom)* exhibit distinct learned behaviors. Under standard non-temporally-coupled attacks, both agents maintain basic body stability, with the `GRAD` agent making an effort to avoid lateral rotations. Notably, WocaR-RL focuses on enhancing robustness in worst-case scenarios, but our experiments reveal its vulnerability to temporally-coupled attacks, leading to a tendency to fall towards one side. In contrast, `GRAD` showcases superior robustness in both non-temporally-coupled and temporally-coupled adversarial settings.

adversary and find an approximate equilibrium. This game-theoretic framework empowers our approach to effectively maximize the agent's performance by adapting to the adversary's strategies.

Our contributions are three-fold. *First*, we propose a novel class of temporally-coupled adversarial attacks to identify the realistic pitfalls of prior threat models as a challenge for existing robust RL methods. *Second*, we introduce a game-theoretic response approach, referred to as `GRAD`. We highlight the significant advantages of GRAD in terms of convergence and policy exploitability. Notably, GRAD demonstrates adaptability to adversaries in both temporally-coupled and non-temporally-coupled settings. Furthermore, GRAD serves as a versatile and flexible solution for adversarial RL, enhancing robustness against diverse types of adversaries.

*Third*, we provide empirical results that demonstrate the effectiveness of our approach in defending against both temporally-coupled and non-temporally coupled adversaries on various attack domains. Figure 1 illustrates how a robustness to temporally-coupled perturbations induces different behavior than robustness to standard perturbations.

## 2 PRELIMINARIES

**Notations and Background.** A Markov decision process (MDP) can be defined as a tuple $\langle \mathcal{S}, \mathcal{A}, \mathcal{P}, \mathcal{R}, \gamma \rangle$, where $\mathcal{S}$ and $\mathcal{A}$ represent the state space and the action space, $\mathcal{R}$ is the reward function: $\mathcal{R} : \mathcal{S} \times \mathcal{A} \to \mathbb{R}$, $\mathcal{P} : \mathcal{S} \times \mathcal{A} \to \Delta(\mathcal{S})$ represents the set of probability distributions over the state space $\mathcal{S}$ and $\gamma \in (0, 1)$ is the discount factor. The agent selects actions based on its policy, $\pi : \mathcal{S} \to \Delta(\mathcal{A})$, which is represented by a function approximator (e.g. a neural network) that is updated during training and fixed during testing. The value function is denoted by $U^\pi(s) := \mathbb{E}_{P,\pi}[\sum_{t=0}^{\infty} \gamma^t R(s_t, a_t) \mid s_0 = s]$, which measures the expected cumulative discounted reward that an agent can obtain from state $s \in \mathcal{S}$ by following policy $\pi$.

**State or Action Adversaries.** State adversary is a type of test-time attacker that perturbs the agent's state observation returned by the environment at each time step and aims to reduce the expected episode reward gained by the agent. While the input to the agent's policy is perturbed, the underlying state in the environment remains unchanged. State adversaries, such as those presented in (Zhang et al., 2020; 2021; Sun et al., 2022), typically consider perturbations on a continuous state space under a certain attack budget $\epsilon$. The attacker perturbs a state $s$ into $\tilde{s} \in \mathcal{B}_\epsilon(s)$, where $\mathcal{B}_\epsilon(s)$ is a $\ell_p$ norm ball centered at $s$ with radius $\epsilon$. Moreover, Action adversaries' goal is to manipulate the behavior of the agent by directly perturbing the action $a$ executed by the agent to $\tilde{a}$ with the probability $\alpha$ as an uncertainty constraint before the environment receives it (altering the output of the agent's policy), causing it to deviate from the optimal policy (Tessler et al., 2019). In this paper, we focus solely on continuous-space perturbations and employ an admissible action perturbation budget as a commonly used $\ell_p$ threat model.

**Algorithm 1** Policy Space Response Oracles (Lanctot et al., 2017)

---

**Result:** Nash Equilibrium
**Input:** Initial population $\Pi^0$
**repeat** {for $t = 0, 1, \ldots$}
    $\pi^r \leftarrow$ NE in game restricted to strategies in $\Pi^t$
    **for** $i \in \{1, 2\}$ **do**
        Find a best response $\beta_i \leftarrow \mathbb{BR}_i(\pi^r_{-i})$
        $\Pi_i^{t+1} \leftarrow \Pi_i^t \cup \{\beta_i\}$
    **end for**
**until** Approximate exploitability is less than or equal to zero
**Return:** $\pi^r$

---

**Problem formulations as a zero-sum game.** We model the game between the agent and the adversary as a two-player zero-sum game that is a tuple $\langle \mathcal{S}, \Pi_a, \Pi_v, \mathcal{P}, \mathcal{R}, \gamma \rangle$, where $\Pi_a$ and $\Pi_v$ denote the sets of policies for the agent and the adversary, respectively. In this framework, both the transition kernels $\mathcal{P}$ and the reward function $\mathcal{R}$ of the victim agent depend on not only its own policy $\pi_a \in \Pi_a$, but also the adversary's policy $\pi_v \in \Pi_v$. The adversary's reward $R(s_t, \bar{a}_t)$ is defined as the negative of the victim agent's reward $R(s_t, a_t)$, reflecting the zero-sum nature of the game. The expected value $u_a^{\pi_a}(h)$ for the agent is the expected sum of future rewards in history $h$ and the robuat RL problem as a two-player zero-sum game has $u_v^{\pi_v}(h) + u_a^{\pi_a}(h) = 0$ for all agent and adversary strategies. A Nash equilibrium (NE) is a strategy profile such that, if all players played their NE strategy, no player could achieve higher value by deviating from it. Formally, $\pi_a^*$ is a NE if $u_a(\pi_a^*) = \max_{\pi_a} u_a(\pi_a, \pi_v^*)$. A best response $\mathbb{BR}$ strategy $\mathbb{BR}_a(\pi_v)$ for the agent $a$ to a strategy $\pi_v$ is a strategy that maximally exploits $\pi_v : \mathbb{BR}_a(\pi_v) = \arg\max_{\pi_a} u_a(\pi_a, \pi_v)$. In this paper, our goal is to converge to the approximate NE for the zero-sum game.

**Double Oracle Algorithm (DO) and Policy Space Response Oracles (PSRO).** Double oracle (McMahan et al., 2003) is an algorithm for finding a Nash Equilibrium (NE) in normal-form games. The algorithm operates by keeping a population of strategies $\Pi^t$ at time $t$. Each iteration, a NE $\pi^{*,t}$ is computed for the game restricted to strategies in $\Pi^t$. Then, a best response $\mathbb{BR}_i(\pi_{-i}^{*,t})$ to this NE is computed for each player $i$ and added to the population, $\Pi_i^{t+1} = \Pi_i^t \cup \{\mathbb{BR}_i(\pi_{-i}^{*,t})\}$ for $i \in \{1, 2\}$. Although in the worst case DO must expand all pure strategies before $\pi^{*,t}$ converges to an NE in the original game, in many games DO terminates early and outperforms alternative methods. An interesting open problem is characterizing games where DO will outperform other methods.

Policy Space Response Oracles (PSRO) (Lanctot et al., 2017; Muller et al., 2019; Feng et al., 2021; McAleer et al., 2022b;a), shown in Algorithm 1 are a method for approximately solving very large games. PSRO maintains a population of reinforcement learning policies and iteratively trains the best response to a mixture of the opponent's population. PSRO is a fundamentally different method than the previously described methods in that in certain games it can be much faster but in other games it can take exponentially long in the worst case.

## 3 ROBUSTNESS TO TEMPORALLY-COUPLED ATTACKS

In this section, we first formally define temporally-coupled attacks. Then, we introduce our algorithm, a game-theoretic response approach for adversarial defense against the proposed attacks.

### 3.1 TEMPORALLY-COUPLED ATTACK

Robust and adversarial RL methods restrict the power of the adversarial by defining a set of admissible perturbations:

**Definition 3.1** ($\epsilon$-Admissible Adversary Perturbations). *An adversarial perturbation $p_t$ is considered admissible in the context of a state adversary if, for a given state $s_t$ at timestep $t$, the perturbed state $\tilde{s}_t$ defined as $\tilde{s}_t = s_t + p_t$ satisfies $\|s_t - \tilde{s}_t\| \leq \epsilon$, where $\epsilon$ is the state budget constraint. Similarly, if $p_t$ is generated by an action adversary, the perturbed action $\tilde{a}_t$ defined as $\tilde{a}_t = a_t + p_t$ should be under the action constraint of $\|a_t - \tilde{a}_t\| \leq \epsilon$.*

While the budget constraint $\epsilon$ is commonly applied in prior adversarial attacks, it may not be applicable in many real-world scenarios where the attacker needs to consider the past perturbations when determining the current perturbations. Specifically, in the temporal dimension, perturbations exhibit a certain degree of correlation. To capture this characteristic, we introduce the concept of temporally-coupled attackers. We propose a temporally-coupled constraint as defined in Definition 3.2, which sets specific limitations on the perturbation at the current timestep based on the previous timestep's perturbation.

**Definition 3.2** ($\bar{\epsilon}$-Temporally-coupled Perturbations). *A temporally-coupled state perturbation $p_t$ is deemed acceptable if it satisfies the temporally-coupled constraint $\bar{\epsilon}$: $\|s_t - \tilde{s}_t - (s_{t+1} - \tilde{s}_{t+1})\| \leq \bar{\epsilon}$ where $\tilde{s}_t$ and $\tilde{s}_{t+1}$ are the perturbed states obtained by adding $p_t$ and $p_{t+1}$ to $s_t$ and $s_{t+1}$, respectively. For action adversaries, the temporally-coupled constraint $\bar{\epsilon}$ is similarly denoted as $\|a_t - \tilde{a}_t - (a_{t+1} - \tilde{a}_{t+1})\| \leq \bar{\epsilon}$, where $\tilde{a}_t$ and $\tilde{a}_{t+1}$ are the perturbed actions.*

We illustrate this definition in Fig. 2. When an adversary is subjected to both of these constraints, it is referred to as a temporally-coupled adversary in this paper. For a temporally-coupled adversary, each timestep's perturbation is restricted within a certain range $\epsilon$, similar to other regular adversarial attacks. However, it is further confined within a smaller range $\bar{\epsilon}$ based on the previous timestep's perturbation. This design offers two significant benefits.

**Figure 2:** *Standard* perturbations and *temporally-coupled* perturbations in a 2d example.

Firstly, it enables the adversary to consider the temporal coupling between perturbations over time. By constraining the perturbations to a smaller range and discouraging drastic changes in direction, the adversary can launch continuous and stronger attacks while preserving a certain degree of stability. Intuitively, if the adversary consistently attacks in one direction, it can be more challenging for the victim to preserve balance and defend effectively compared to when the perturbations alternate between the left and right directions.

Then, the temporally-coupled constraint also enables the adversary to efficiently discover the optimal attack strategy by narrowing down the range of choices for each timestep's perturbation. Reducing the search space does not necessarily weaken the adversary; in fact, it can potentially make the adversary stronger if the optimal attack lies within the temporally-determined search space, which is supported by our empirical results. By constraining the adversary to a more focused exploration of attack strategies, the temporally-coupled constraint facilitates the discovery and exploitation of more effective and targeted adversarial tactics that exhibit less variation at consecutive timesteps. This characteristic enhances the adversary's ability to launch consistent and potent attacks.

Practically, it is crucial to carefully determine $\bar{\epsilon}$ to guarantee that this additional temporally-coupled constraint does not impede the performance of attacks but rather amplifies their effectiveness. The effectiveness of different choices for $\bar{\epsilon}$ was empirically evaluated in our empirical studies, highlighting the benefits it brings to adversarial learning. Temporally-coupled perturbations represent a novel case to challenge the existing methods. Our empirical experiments demonstrate that, even with the introduction of temporally-coupled constraints, these perturbations can have a notable impact on existing robust models, showcasing the need for addressing such scenarios in robust RL.

## 3.2 GRAD: GAME-THEORETIC APPROACH FOR ADVERSARIAL DEFENSE

Building upon prior robust RL methods, we develop a robust RL algorithm for the temporally-coupled setting. Our resulting method uses tools from game theory to enhance robustness against adversaries with different settings including non-temporally-coupled and temporally-coupled constraints.

In our Game-theoretic Response approach for Adversarial Defense ( GRAD) framework as a modification of PSRO (Lanctot et al., 2017), an agent and a temporally-coupled adversary are trained as part of a two-player game. They play against each other and update their policies in response to each other's policies. The adversary is modeled as a separate agent who attempts to maximize the impact of attacks on the original agent's performance and whose action space is constrained by both $\epsilon$ and $\bar{\epsilon}$. Our method adapts the $\epsilon$-budget assumption from prior work (Liang et al., 2022) to handle temporally-coupled constraints. Meanwhile, the original agent's objective function is based

---

**Algorithm 2** Game-theoretic Response approach for Adversarial Defense ( GRAD)

---

**Input:** Initial policy sets for the agent and adversary $\Pi : \{\Pi_a, \Pi_v\}$
Compute expected utilities as empirical payoff matrix $U^\Pi$ for each joint $\pi : \{\pi_a, \pi_v\} \in \Pi$
Compute meta-Nash equilibrium $\sigma_a$ and $\sigma_v$ over policy sets $(\Pi_a, \Pi_v)$
**for** epoch in $\{1, 2, \ldots\}$ **do**
    **for** many iterations $N_{\pi_a}$ **do**
        Sample the adversary policy $\pi_v \sim \sigma_v$
        Train $\pi_a'$ with trajectories against the fixed adversary $\pi_v$: $\mathcal{D}_{\pi_a'} := \{(\hat{s}_t^k, a_t^k, r_t^k, \hat{s}_{t+1}^k)\}\big|_{k=1}^{B}$
        (when the fixed adversary only attacks the action space, $\hat{s}_t = s_t$.)
    **end for**
    $\Pi_a = \Pi_a \cup \{\pi_a'\}$
    **for** many iterations $N_{\pi_v}$ **do**
        Sample the agent policy $\pi_a \sim \sigma_a$
        Train the adversary policy $\pi_v'$ with trajectories: $\mathcal{D}_{\pi_v'} := \{(s_t^k, \bar{a}_t^k, -r_t^k, s_{t+1}^k)\}\big|_{k=1}^{B}$
        ($\pi_v'$ applies attacks to the fixed victim agent $\pi_a$ based on $\bar{a}_t$ using different methods)
    **end for**
    $\Pi_v = \Pi_v \cup \{\pi_v'\}$
    Compute missing entries in $U^\Pi$ from $\Pi$
    Compute new meta strategies $\sigma_a$ and $\sigma_v$ from $U^\Pi$
**end for**
**Return:** current meta Nash equilibrium on whole population $\sigma_a$ and $\sigma_v$

---

on the reward obtained from the environment, taking into account the perturbations imposed by the adversary. The process continues until an approximate equilibrium is reached, at which point the original agent is considered to be robust to the attacks learned by the adversary. We show our full algorithm in Algorithm 2.

Under some assumptions (see Appendix A for details), GRAD convergence to an approximate Nash Equilibrium (NE):

**Proposition 3.3.** *For a finite-horizon MDP with a fixed number of discrete actions, GRAD converges to an approximate Nash Equilibrium (NE) of the two-player zero-sum adversarial game.*

In GRAD, both the agent and the adversary have two policy sets. During each training epoch, the agent aims to find an approximate best response to the fixed adversary, and vice versa for the adversary. This iterative process promotes the emergence of stable and robust policies. After each epoch, the new trained policies are added to the respective policy sets, which allows for a more thorough exploration of the policy space.

For different types of attackers, the agent generates different trajectories while training against a fixed attacker. If the attacker only targets the state, then the agent's training data will consist of the altered state $\hat{s}$ after adding the perturbations from the fixed attacker. If the attacker targets the agent's action, the agent's policy output $a$ will be altered as $\hat{a}$ by the attacker, even if the agent receives the correct state $s$ during training. As for the adversary's training, after defining the adversary's attack method and policy model, the adversary applies attacks to the fixed agent and collects the trajectory data and the negative reward to train the adversary. The novelty of GRAD lies in its scalability and adaptability in robust RL which converges to the approximate equilibrium without only considering certain adversarial scenarios. Our work formulates the robust RL objective as a zero-sum game and demonstrates the efficacy of game-theoretic RL in tackling this objective, rather than being solely reliant on the specific game-theoretic RL algorithm of PSRO or focusing on defense against specific types of attackers. GRAD provides a more versatile and adaptive solution.

As in Definition 3.2, which is actually a generalization of the original attack space. We have $\|s_t - \tilde{s}_t - (s_{t+1} - \tilde{s}_{t+1})\| \leq \|s_t - \tilde{s}_t\| + \|s_{t+1} - \tilde{s}_{t+1}\| \leq 2\epsilon$, so when $\bar{\epsilon} > 2\epsilon$, it converges to the non-coupled attack scenario. Consequently, our defense strategy is not specific to a narrow attack set In the next section, we empirically demonstrate that our approach exhibits superior and comprehensive robustness, which is capable of adapting to various attack scenarios and effectively countering different types of adversaries on continuous control tasks.

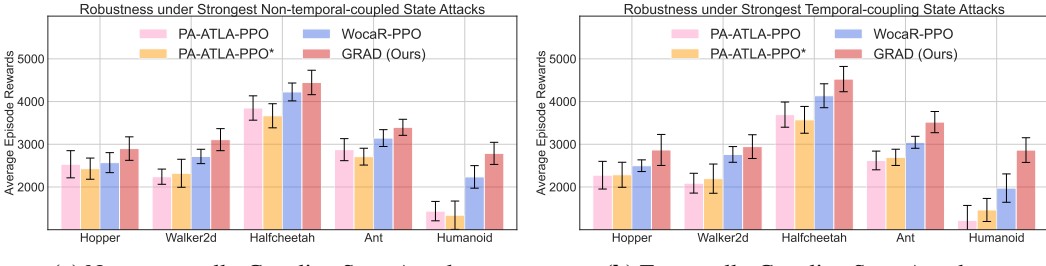

**(a)** Non-temporally-Coupling State Attacks      **(b)** Temporally-Coupling State Attacks

**Figure 3:** Average episode rewards ± standard deviation over 100 episodes under the strongest non-temporally-coupling and temporally-coupling state attacks for state robust baselines and `GRAD` on five control tasks.

## 4 EXPERIMENTS

In our experiments, we investigate various types of attackers on different attack domains including state perturbations, action perturbations, model uncertainty and mixed perturbations. We will study a diverse set of attack and compare with state-of-the-art baselines.

**Experiment setup.** Our experiments are conducted on five various and challenging MuJoCo environments: Hopper, Walker2d, Halfcheetah, Ant, and Humanoid, all using the v2 version of MuJoCo. We use the Proximal Policy Optimization (PPO) algorithm as the policy optimizer for `GRAD` training. For attack constraint $\epsilon$, we use the commonly adopted values $\epsilon$ for each environment. We set the temporally-coupled constraint $\bar{\epsilon} = \epsilon/5$ (with minor adjustments in some environments). Ablation experiments study the choice ofOther choices of $\bar{\epsilon}$ will be further discussed in the ablation studies. Our experiments are conducted on five various and challenging MuJoCo environments: Hopper, Walker2d, Halfcheetah, Ant, and Humanoid, all using the v2 version of MuJoCo. We use the Proximal Policy Optimization (PPO) algorithm as the policy optimizer for `GRAD` training. For attack constraint $\epsilon$, we use the commonly adopted values $\epsilon$ for each environment. We set the temporally-coupled constraint $\bar{\epsilon} = \epsilon/5$ (with minor adjustments in some environments). Ablation experiments study the choice of $\bar{\epsilon}$.

We report the average test episodic rewards both under no attack and against the strongest adversarial attacks to reflect both the natural performance and robustness of trained agents, by training adversaries targeting the trained agents from scratch. For reproducibility, we train each agent configuration with 10 seeds and report the one with the median robust performance, rather than the best one. More implementation details are in Appendix C.1.

**Case I: Robustness against state perturbations.** In this experiment, our focus is on evaluating the robustness of our methods against state adversaries that perturb the states received by the agent. Among the alternating training (Zhang et al., 2021; Sun et al., 2022) methods, PA-ATLA-PPO is the most robust, which trains with the standard strongest PA-AD attacker. As a modification, we train PA-ATLA-PPO* with a temporally-coupled PA-AD attacker. WocaR-PPO (Liang et al., 2022) is the state-of-the-art defense method against state adversaries. Our `GRAD` method utilizes the temporally-coupled PA-AD attacker for training. Figure 3 presents the performance of baseline and `GRAD` under both non-temporally-coupled and temporally-coupled state perturbations.

Despite being trained to handle temporally-coupled adversaries, our method also demonstrates strong performance in the non-robust ("natural") setting, especially on the high-dimensional Humanoid task. Under our temporally-coupled attacks, the average performance of `GRAD` is 45% higher than the strongest baseline.

**Case II: Robustness against action uncertainty.** Beyond assessing the susceptibility of `GRAD` to state attacks, we also investigate its robustness against action uncertainty, where the agent intends to execute an action but ultimately takes a different action than anticipated. We scrutinize two specific forms of action uncertainty, as outlined in prior work (Tessler et al., 2019). The first one is action perturbations, introduced by an action adversary, which strategically adds noise to the agent's intended action. The second scenario revolves around model uncertainty, where, with a probability denoted as $\alpha$, an alternative action replaces the originally planned action output by the agent. These

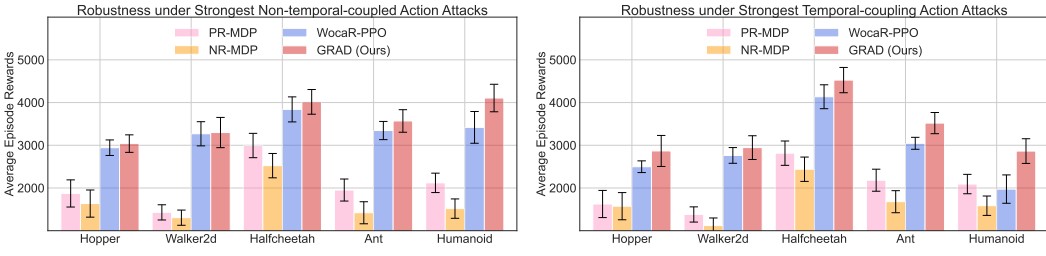

**(a)** Non-temporally-Coupling Action Attacks        **(b)** Temporally-Coupling Action Attacks

**Figure 4:** Average episode rewards ± standard deviation over 100 episodes for `GRAD` and action robust models against the strongest non-temporally-coupled and temporally-coupled action perturbations on five MuJoCo tasks.

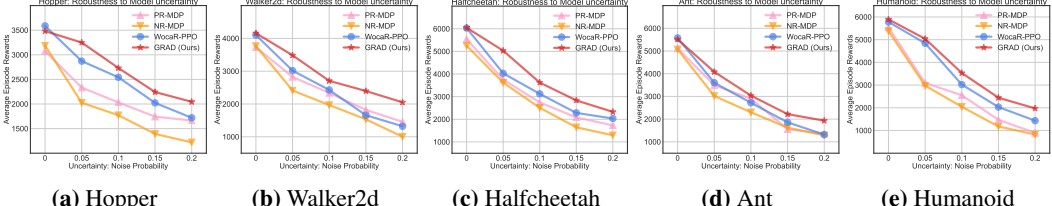

**(a)** Hopper        **(b)** Walker2d        **(c)** Halfcheetah        **(d)** Ant        **(e)** Humanoid

**Figure 5:** Robustness to Model Uncertainty Across Various $\alpha$ Values. The noisy probability $\alpha$ represents the likelihood of a randomly sampled noise replacing the initially selected action.

scenarios closely parallel real-world control situations, such as dealing with mass uncertainty (e.g., when a robot's weight changes) or facing sudden, substantial external forces (e.g., when an external force unexpectedly pushes a robot).

In our baseline comparisons, we include PR-MDP and NR-MDP (Tessler et al., 2019), which are robust to action noise and model uncertainty. We also incorporate WocaR-PPO into our baseline evaluations. We train `GRAD` using a temporally-coupled action adversary and evaluate its robustness in both action perturbation and model uncertainty scenarios.

**Action Perturbations.** To obtain a stronger evasion action perturbation rather than OU noise and parameter noise, we are the first to train an RL-based action adversary following the trajectory outlined in Algorithm 2. This strategy aims to showcase the worst-case performance of our robust agents under action perturbations. For evaluation, we train both temporally-coupled and non-temporally-coupled action adversaries for each robust model. In Figure 4, we present the exceptional performance of `GRAD` against standard and temporally-coupling action perturbations. `GRAD` demonstrates a high degree of robustness. For example, on the Humanoid task it outperforms the baselines by a 17% margin for standard attacks and by a 40% advantage against temporally-coupling action attacks. These results provide evidence of `GRAD`'s defense mechanism against various types of adversarial attacks in the action space.

**Model Uncertainty.** To evaluate robustness under model uncertainty, we consider a range of noise probabilities denoted as $\alpha$ in the range of [0, 0.05, 0.1, 0.15, 0.2]. These values represent the probability of a randomly generated noise replacing the action selected by the victim agent. As depicted in Figure 5, `GRAD` exhibits superior robustness compared to action-robust baselines across a spectrum of $\alpha$ uncertainty value without explicit exposure to model uncertainty noises during training.

**Case III: Robustness against mixed adversaries.** In prior works, adversarial attacks typically focused on perturbing either the agent's observations or introducing noise to the action space. However, in real-world scenarios, agents may encounter both types of attacks simultaneously. To address this challenge, we propose a mixed adversary, which allows the adversary to perturb the agent's state and action at each time step. We employ alternating training to create a baseline as Mixed-ATLA using this mixed adversary type. Our `GRAD` model and Mixed-ATLA are trained with temporally-coupled mixed attackers. The detailed algorithm for the mixed adversary is provided in Appendix 5.

Our results in Figure 6 indicate that the combination of two different forms of attacks can target robust agents in most scenarios, providing strong evidence of their robustness. `GRAD` outperforms other

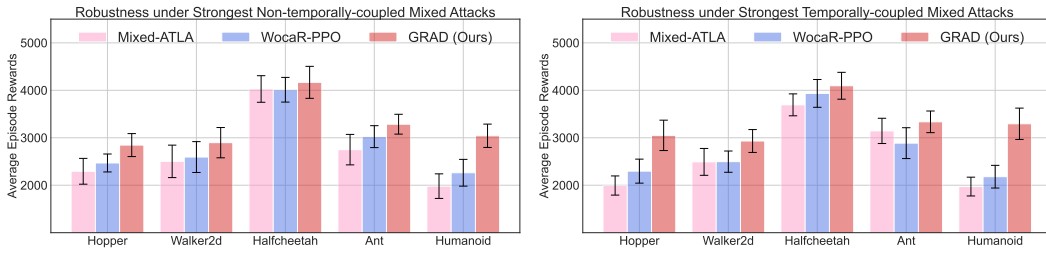

**(a)** Non-temporally-Coupling Mixed Attacks    **(b)** Temporally-Coupling Mixed Attacks

**Figure 6:** Average episode rewards ± standard deviation of GRAD and baselines over 100 episodes under the strongest non-temporally-coupling and temporally-coupling mixed attacks.

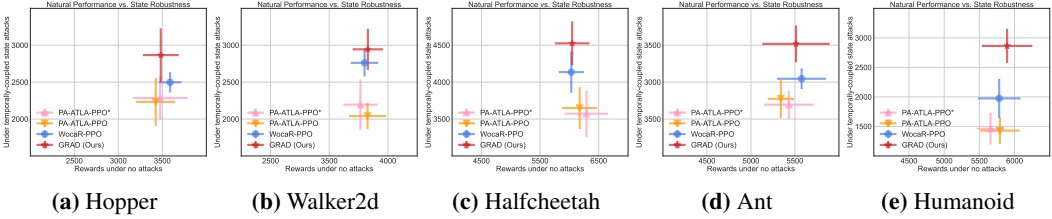

**(a)** Hopper    **(b)** Walker2d    **(c)** Halfcheetah    **(d)** Ant    **(e)** Humanoid

**Figure 7:** GRAD achieves higher returns in the robust setting without sacrificing performance in the non-robust ("natural") setting.

methods in all five environments against non-temporally-coupled mixed adversaries, with a margin of over 20% in the Humanoid environment. Moreover, when defending against temporally-coupled mixed attacks, GRAD outperforms baselines by 30% in multiple environments, with a minimum improvement of 10%.

**Natural Performance.**    We also evaluate the natural performance of GRAD and the baselines, as shown in Figure 7, which compares natural rewards vs. rewards under the strongest temporally-coupled attacks. It is evident that while achieving robustness, GRAD maintains a comparable natural performance with the baselines; the agent's performance does not degrade significantly in environments without adversaries. The natural performance comparing GRAD with action-robust models can be found in Appendix C.5.

**Ablation studies for temporally-coupled constraint $\bar{\epsilon}$.**    As defined in our framework, the temporally-coupled constraint $\bar{\epsilon}$ limits the perturbations within a range that varies between timesteps. When $\bar{\epsilon}$ is set too large, the constraint becomes ineffective, resembling a standard attacker. Conversely, setting $\bar{\epsilon}$ close to zero overly restricts perturbations, leading to a decline in attack performance. An appropriate value for $\bar{\epsilon}$ is critical for effective temporally-coupled attacks. Figure 8 illustrates the performance of robust models against temporally-coupled state attackers trained with different maximum $\bar{\epsilon}$. For WocaR-PPO, the temporally-coupled attacker achieves good performance when the values of $\bar{\epsilon}$ are set to 0.02. As the $\bar{\epsilon}$ values increase and the temporally-coupled constraint weakens, the agent's performance improves, indicating a decrease in the adversary's attack effectiveness. In the case of

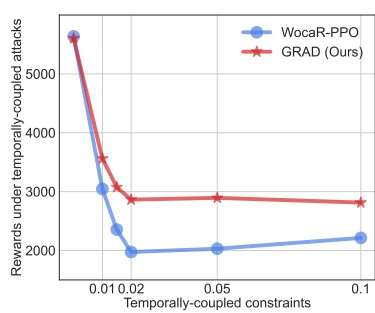

**Figure 8:** Ablated studies for $\bar{\epsilon}$.

GRAD agents, they consistently maintain robust performance as the $\bar{\epsilon}$ values become larger. This observation highlights the impact of temporal coupling on the vulnerability of robust baselines to such attacks. In contrast, GRAD agents consistently demonstrate robustness against these attacks.

## 5    RELATED WORK

**Robust RL against adversarial perturbations.** Existing defense approaches for RL agents are primarily designed to counter adversarial perturbations in state observations. These methods encompass a wide range of strategies, including regularization techniques (Zhang et al., 2020; Shen et al.,

2020; Oikarinen et al., 2021), attack-driven approaches involving weak or strong gradient-based attacks (Kos & Song, 2017; Behzadan & Munir, 2017; Mandlekar et al., 2017; Pattanaik et al., 2018; Franzmeyer et al., 2022; Vinitsky et al., 2020), RL-based alternating training methods (Zhang et al., 2021; Sun et al., 2022), and worst-case motivated methods (Liang et al., 2022). Furthermore, there is a line of research that delves into providing *theoretical guarantees* for adversarial defenses in RL (Lütjens et al., 2020; Oikarinen et al., 2021; Fischer et al., 2019; Kumar et al., 2022; Wu et al., 2022; Sun et al., 2023), exploring a variety of settings and scenarios where these defenses can be effectively applied.Aversarial attacks can take various forms. For instance, perturbations can affect the actions executed by the agent (Pan et al., 2022; Tessler et al., 2019; Lanier et al., 2022; Lee et al., 2020). Additionally, the study of adversarial multi-agent games has also received attention (Gleave et al., 2020; Pinto et al., 2017).

**Robust Markov decision process and safe RL.** There are several lines of work that study RL under safety/risk constraints (Heger, 1994; Gaskett, 2003; García & Fernández, 2015; Bechtle et al., 2020; Thomas et al., 2021) or under intrinsic uncertainty of environment dynamics (Lim et al., 2013; Mankowitz et al., 2020). In particular, several works discuss coupled or non-rectangular uncertainty sets, which allow less conservative and more efficient robust policy learning by incorporating realistic conditions that naturally arise in practice. Mannor et al. (2012) propose to model coupled uncertain parameters based on the intuition that the total number of states with deviated parameters will be small. Mannor et al. (2016) identify "k-rectangular" uncertainty sets defined by the cardinality of possible conditional projections of uncertainty sets, which can lead to more tractable solutions. Another recent work (Goyal & Grand-Clement, 2023) proposes to model the environment uncertainty with factor matrix uncertainty sets, which can efficiently compute a robust policies.

**Two-player zero-sum games.** There are a number of related deep reinforcement learning methods for two-player zero-sum games. CFR-based techniques such as Deep CFR (Brown et al., 2019), DREAM (Steinberger et al., 2020), and ESCHER (McAleer et al., 2023), use deep reinforcement learning to approximate CFR. Policy-gradient techniques such as NeuRD (Hennes et al., 2020), Friction-FoReL (Perolat et al., 2021; 2022), and MMD (Sokota et al., 2022), approximate Nash equilibrium via modified actor-critic algorithms. Our robust RL approach takes the double oracle techniques such as PSRO (Lanctot et al., 2017) as the backbone. PSRO-based algorithms have been shown to outperform the previously-mentioned algorithms in certain games (McAleer et al., 2021).

A more detailed discussion of related works in robust RL and game-theoretic RL are in Appendix B.

## 6 Conclusion and Discussion

Motivated by the perturbations that arise in real world scenarios, we introduce a new attack model for studying deep RL models. Since existing robust RL methods usually focus on a traditional threat model that perturbs state observations or actions arbitrarily within an $L_p$ norm ball, they become too conservative and can fail to perform a good defense under the temporally-coupled attacks. In contrast, we propose a game-theoretical response approach GRAD, which finds the best response against attacks with various constraints including temporally-coupled ones. Experiments across a range of continuous control tasks underscore the good performance of our approach over previous robust RL methods for both non-temporally-coupled attacks and temporally-coupled attacks across diverse attack domains.

**Limitations.** The current PSRO-based approach may require several iterations to converge to the best response, which can pose limitations when computational resources are constrained. We leverage distributed RL tools to expedite the training of RL agents within GRAD, enabling efficient learning of the best response. Detailed computational cost analysis can be found in Appendix C.6.

Regarding scalability concerns, we have demonstrated the GRAD in addressing robust RL problems on high-dimensional tasks. In principle, alternative game-theoretic algorithms (Perolat et al., 2022), known for their practical efficiency, can be considered for defense in different game scenarios. As part of our future research directions, we plan to explore methods to further enhance the scalability of GRAD. This exploration may involve harnessing parallel training techniques and drawing insights from other scalable PSRO approaches (McAleer et al., 2020; Lanctot et al., 2017). Additionally, we aim to extend the applicability of our method to pixel-based RL scenarios and real-world situations with increased practicality and complexity.

## ACKNOWLEDGEMENTS

Liang, Zheng, Liu and Huang are supported by National Science Foundation NSF-IIS-2147276 FAI, DOD-ONR-Office of Naval Research under award number N00014-22-1-2335, DOD-AFOSR-Air Force Office of Scientific Research under award number FA9550-23-1-0048, DOD-DARPA-Defense Advanced Research Projects Agency Guaranteeing AI Robustness against Deception (GARD) HR00112020007, Adobe, Capital One and JP Morgan faculty fellowships. McAleer is funded by NSF grant #2127309 to the Computing Research Association for the CIFellows 2021 Project.

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

# A  PROOF OF PROPOSITION 3.3

*Proof.* The *exploitability* $e(\pi)$ of a strategy profile $\pi$ is defined as

$$e(\pi) = \sum_{i \in \mathcal{N}} \max_{\pi_i'} v_i(\pi_i', \pi_{-i}).$$

A *best response (BR)* strategy $\mathbb{BR}_i(\pi_{-i})$ for player $i$ to a strategy $\pi_{-i}$ is a strategy that maximally exploits $\pi_{-i}$:

$$\mathbb{BR}_i(\pi_{-i}) = \arg \max_{\pi_i} v_i(\pi_i, \pi_{-i}).$$

An $\epsilon$-*Nash equilibrium ($\epsilon$-NE)* is a strategy profile $\pi$ in which, for each player $i$, $\pi_i$ is an $\epsilon$-BR to $\pi_{-i}$.

We can define the *approximate exploitability* of the pair of meta-Nash equilibrium strategies for the agent and the adversary $(\sigma_a, \sigma_v)$ as the sum of the expected reward their opponent approximate best responses achieve against them:

$$\hat{e}(\sigma_a, \sigma_v) = v_a(\pi_a', \sigma_v) + v_v(\pi_v', \sigma_u),$$

where $v_i(\pi_i', \sigma_{-i})$ denotes the expected value of a player's approximate best response vs. the opponent meta-NE.

Now assume that each approximate best response is within $\frac{\epsilon}{4}$ of the optimal best response. Then

$$v_i(\pi_i', \sigma_{-i}) \geq v_i(\mathbb{BR}_i(\sigma_{-i}), \sigma_{-i}) - \frac{\epsilon}{4}.$$

As a result, upon convergence, when the approximate exploitability $\hat{e}(\sigma_a, \sigma_v)$ is less than $\frac{\epsilon}{2}$, then the exploitability of the pair of meta-Nash equilibrium strategies for the agent and the adversary $(\sigma_a, \sigma_v)$ is less than $\epsilon$, and the pair of strategies are in an $\epsilon$-approximate Nash equilibrium.

Every epoch where GRAD does not converge to an approximate equilibrium, it must add a unique deterministic policy to the population for either the agent or the adversary because if both players added policies already included in their populations, those policies would not be approximate best responses. Given that the MDP has a finite horizon and operates in a discrete action space, there exists only a finite set of deterministic policies that can be added to the populations $\Pi_a$ and $\Pi_v$. Since the meta-Nash equilibrium over all possible deterministic policies is equivalent to the Nash equilibrium of the original game, in the worst case where all possible deterministic policies are added, the algorithm will terminate at an approximate Nash equilibrium.

# B  ADDITIONAL RELATED WORK

**Robust RL against adversarial perturbations.** *Regularization-based methods* (Zhang et al., 2020; Shen et al., 2020; Oikarinen et al., 2021) enforce the policy to have similar outputs under similar inputs, which can achieve certifiable performance for visual-input RL (Xu et al., 2023a) on Atari games. However, in continuous control tasks, these methods may not reliably improve the worst-case performance. Recent work by Korkmaz (2021) points out that these adversarially trained models may still be sensible to new perturbations. *Attack-driven methods* train DRL agents with adversarial examples. Some early works (Kos & Song, 2017; Behzadan & Munir, 2017; Mandlekar et al., 2017; Pattanaik et al., 2018; Franzmeyer et al., 2022; Vinitsky et al., 2020) apply weak or strong gradient-based attacks on state observations to train RL agents against adversarial perturbations. Zhang et al. (2021) and Sun et al. (2022) propose to alternately train an RL agent and a strong RL adversary, namely ATLA, which significantly improves the policy robustness against rectangle state perturbations. A recent work by Liang et al. (2022) introduces a more principled adversarial training framework that does not explicitly learn the adversary, and both the efficiency and robustness of RL agents are boosted.

Additionally, a significant body of research has delved into providing *theoretical guarantees* for adversarial defenses in RL (Lütjens et al., 2020; Oikarinen et al., 2021; Fischer et al., 2019; Kumar et al., 2022; Wu et al., 2022; Sun et al., 2023), exploring various settings and scenarios. Robust RL faces challenges under model uncertainty in prior works (Iyengar, 2005; Nilim & Ghaoui, 2005; Xu et al., 2023b). The main goal of GRAD is to address the adversarial RL problem with an adversary that is adaptive to the agent's policy. Works like Tessler et al. (2019) on action perturbations and

Zhou et al. (2023) on model mismatch uncertainty, are hard to defend against the strongest adversarial perturbations and only empirically evaluated on uncertainty sets. This vulnerability arises due to the inherent difficulty in estimating the long-term worst-case value under adaptive adversaries. Distributional robust optimization (DRO)(Rahimian & Mehrotra, 2019) is also challenging to apply to this challenging problem, especially against state adversaries in the high-dimensional state space. At the same time, adversarial training methods also struggle to effectively deal with model uncertainty or model mismatch problems. However, GRAD, leveraging game-theoretic methods, demonstrates robustness against adversarial perturbations and model uncertainty, as a more effective and general solution for high-dimensional tasks.

**Robust RL formulated as a zero-shot game.** Considering robust RL formulated as zero-sum games, several notable contributions have emerged. Pinto et al. (2017) proposed robust adversarial reinforcement learning (RARL), introducing the concept of training an agent in the presence of a destabilizing adversary through a zero-sum minimax objective function. Huang et al. (2022) extended this paradigm with RRL-Stack, a hierarchical formulation of robust RL using a general-sum Stackelberg game model. Tessler et al. (2019) addressed action uncertainty by framing the action perturbation problem as a zero-sum game. GRAD distinguishes itself from conventional robust RL approaches by achieving approximate equilibriums on an adversary policy set. While existing works often focus on specific adversaries or worst-case scenarios, limiting their adaptability, GRAD does not specifically target certain adversaries, which makes it adaptable to various adversaries, including both temporally-coupled and non-temporally-coupled adversarial settings. Moreover, GRAD's broad applicability extends to diverse attack domains, presenting a more practical and scalable solution for robust RL compared to prior works.

**Game-Theoretic Reinforcement Learning.** Superhuman performance in two-player games usually involves two components: the first focuses on finding a model-free blueprint strategy, which is the setting we focus on in this paper. The second component improves this blueprint online via model-based subgame solving and search (Burch et al., 2014; Moravcik et al., 2016; Brown et al., 2018; 2020; Brown & Sandholm, 2017b; Schmid et al., 2021). This combination of blueprint strategies with subgame solving has led to state-of-the-art performance in Go (Silver et al., 2017), Poker (Brown & Sandholm, 2017a; 2018; Moravčík et al., 2017), Diplomacy (Gray et al., 2020), and The Resistance: Avalon (Serrino et al., 2019). Methods that only use a blueprint have achieved state-of-the-art performance on Starcraft (Vinyals et al., 2019), Gran Turismo (Wurman et al., 2022), DouDizhu (Zha et al., 2021), Mahjohng (Li et al., 2020), and Stratego (McAleer et al., 2020; Perolat et al., 2022). In the rest of this section we focus on other model-free methods for finding blueprints.

Deep CFR (Brown et al., 2019; Steinberger, 2019) is a general method that trains a neural network on a buffer of counterfactual values. However, Deep CFR uses external sampling, which may be impractical for games with a large branching factor, such as Stratego and Barrage Stratego. DREAM (Steinberger et al., 2020) and ARMAC (Gruslys et al., 2020) are model-free regret-based deep learning approaches. ReCFR (Liu et al., 2022) proposes a bootstrap method for estimating cumulative regrets with neural networks. ESCHER (McAleer et al., 2023) removes the importance sampling term of Deep CFR and shows that doing so allows scaling to large games.

Neural Fictitious Self-Play (NFSP) (Heinrich & Silver, 2016) approximates fictitious play by progressively training the best response against an average of all past opponent policies using reinforcement learning. The average policy converges to an approximate Nash equilibrium in two-player zero-sum games.

There is an emerging literature connecting reinforcement learning to game theory. QPG (Srinivasan et al., 2018) shows that state-conditioned $Q$-values are related to counterfactual values by a reach weighted term summed over all histories in an infostate and proposes an actor-critic algorithm that empirically converges to an NE when the learning rate is annealed. NeuRD (Hennes et al., 2020), and F-FoReL (Perolat et al., 2021) approximate replicator dynamics and follow the regularized leader, respectively, with policy gradients. Actor Critic Hedge (ACH) (Fu et al., 2022) is similar to NeuRD but uses an information set based value function. All of these policy-gradient methods do not have a theory proving that they converge with high probability in extensive form games when sampling trajectories from the policy. In practice, they often perform worse than NFSP and DREAM on small games but remain promising approaches for scaling to large games (Perolat et al., 2022).

---

**Algorithm 3** Action Adversary (AC-AD)

---

**Input:** Initialization of action adversary policy $v$; victim policy $\pi$, initial state $s_0$
**for** $t = 0, 1, 2, \ldots$ **do**
    adversary $v$ samples an action perturbations $\widehat{a}_t \sim \nu(\cdot|s_t)$,
    victim policy $\pi$ outputs action $a_t \sim \pi(\cdot|s_t)$
    the environment receives $\tilde{a}_t = a_t + \widehat{a}_t$, returns $s_{t+1}$ and $r_t$
    adversary saves $(s_t, \widehat{a}_t, -r_t, s_{t+1})$ to the adversary buffer
    adversary updates its policy $v$
**end for**

---

## C  EXPERIMENT DETAILS AND ADDITIONAL RESULTS

### C.1  IMPLEMENTATION DETAILS

We provide detailed implementation information for our proposed method (`GRAD`) and baselines.

**Training Steps**  For `GRAD`, we specify the number of training steps required for different environments. In the Hopper, Walker2d, and Halfcheetah environments, we train for 10 million steps. In the Ant and Humanoid environments, we extend the training duration to 20 million steps. For the ATLA baselines, we train for 2 million steps and 10 million steps in environments of varying difficulty.

**Network Structure**  Our algorithm (`GRAD`) adopts the same PPO network structure as the ATLA baselines to maintain consistency. The network comprises a single-layer LSTM with 64 hidden neurons. Additionally, an input embedding layer is employed to project the state dimension to 64, and an output layer is used to project 64 to the output dimension. Both the agents and the adversaries use the same policy and value networks to facilitate training and evaluation. Furthermore, the network architecture for the best response and meta Nash remains consistent with the aforementioned configuration.

**Schedule of $\epsilon$ and $\bar{\epsilon}$**  During the training process, we gradually increase the values of $\epsilon$ and $\bar{\epsilon}$ from 0 to their respective target maximum values. This incremental adjustment occurs over the first half of the training steps. We reference the attack budget $\epsilon$ used in other baselines for the corresponding environments. This ensures consistency and allows for a fair comparison with existing methods. The target value of $\bar{\epsilon}$ is determined based on the adversary's training results, which is set as $\epsilon/5$. In some smaller dimensional environments, $\bar{\epsilon}$ can be set to $\epsilon/10$. We have observed that the final performance of the trained robust models does not differ by more than 5% when using these values for $\bar{\epsilon}$.

**Observation and Reward Normalization**  To ensure consistency with PPO implementation and maintain comparability across different codebases, we apply observation and reward normalization. Normalization helps to standardize the input observations and rewards, enhancing the stability and convergence of the training process. We have verified the performance of vanilla PPO on different implementations, and the results align closely with our implementation of `GRAD` based on Ray rllib.

**Hyperparameter Selection**  Hyperparameters such as learning rate, entropy bonus coefficient, and other PPO-specific parameters are crucial for achieving optimal performance. Referring to the results obtained from vanilla PPO and the ATLA baselines as references, a small-scale grid search is conducted to fine-tune the hyperparameters specific to `GRAD`. Because of the significant training time and cost associated with `GRAD`, we initially perform a simplified parameter selection using the Inverted Pendulum as a test environment.

### C.2  ADVERSARIES IN EXPERIMENTS

**State Adversaries**  Aimed to introduce the attack methods utilized during training and testing in our experiments. When it comes to state adversaries, PA-AD as Alogrithm 4 stands out as the strongest attack compared to other state attacks. Therefore, we report the best state attack rewards under PA-AD attacks.

**Action Adversaries**  In terms of action adversaries, an RL-based action adversary as Alogrithm 3 can inflict more severe damage on agents' rewards compared to OU noise and parameter noise in (Tessler et al., 2019).

**Mixed Adversaries** When dealing with mixed adversaries capable of perturbing both state and action spaces, it becomes crucial to design the action space for the adversary. In Algorithm 5, we extend the idea of PA-AD (Sun et al., 2022), which learns a policy perturbation direction to generate perturbations. In our case, the mixed adversary director only needs to learn the policy perturbation direction $\hat{d}_t$. For various attack domains, the actor functions then translate the direction $\hat{d}_t$ into state or action perturbations. This design approach ensures that our mixed adversary doesn't increase the complexity of adversary training, as it deploys mixed perturbations using different actor functions as required by distinct attack domains.

---

**Algorithm 4** Policy Adversarial Actor Director (PA-AD)

---

**Input:** Initialization of adversary director's policy $v$; victim policy $\pi$, the actor function $g$ for the state space $\mathcal{S}$, initial state $s_0$
**for** $t = 0, 1, 2, \ldots$ **do**
  *Director* $v$ samples a policy perturbing direction and perturbed choice, $\hat{a}_t \sim \nu(\cdot|s_t)$
  *Actor* perturbs $s_t$ to $\tilde{s}_t = g(\hat{a}_t, s_t)$
  *Victim* takes action $a_t \sim \pi(\cdot|\tilde{s}_t)$, proceeds to $s_{t+1}$, receives $r_t$
  *Director* saves $(s_t, \hat{a}_t, -r_t, s_{t+1})$ to the adversary buffer
  *Director* updates its policy $v$ using any RL algorithms
**end for**

---

**Algorithm 5** Mixed Adversary

---

**Input:** Initialization of adversary director's policy $v$; victim policy $\pi$, the actor function $g_s$ for the state space $\mathcal{S}$ and $g_a$ for the action space $\mathcal{A}$, initial state $s_0$
**for** $t = 0, 1, 2, \ldots$ **do**
  *Director* $v$ samples a policy perturbing direction $\hat{d}_t \sim \nu(\cdot|s_t)$.
  *Actor* perturbs $s_t$ to $\tilde{s}_t = g_s(\hat{d}_t, s_t)$
  *Victim* takes action $a_t \sim \pi(\cdot|\tilde{s}_t)$, proceeds to $s_{t+1}$, receives $r_t$
  victim policy outputs action $a_t \sim \pi(\cdot|s_t)$
  *Actor* perturbs $a_t$ to $\tilde{a}_t = g_a(\hat{d}_t, a_t)$
  The environment receives $\tilde{a}_t$, returns $s_{t+1}$ and $r_t$
  *Director* saves $(s_t, \hat{a}_t, -r_t, s_{t+1})$ to the adversary buffer
  *Director* updates its policy $v$ using any RL algorithms
**end for**

---

**Transition Adversaries.** In addition to addressing adversarial perturbations, we extend the evaluation of `GRAD` to consider transition uncertainty, mitigating the mismatch problem between the training simulator and the testing environment. Robustness under transition uncertainty is crucial for real-world applicability. To assess this aspect, experiments are conducted on perturbed MuJoCo environments (Hopper, Walker2d, and HalfCheetah) by modifying their physical parameters ('leg_joint_stiffness' value: 30, 'foot_joint_stiffness' value: 30, and bound on 'back_actuator_range': 0.5) following the protocol established by Zhou et al. (2023). Comparative evaluations are performed against robust natural actor-critic (RNAC)(Zhou et al., 2023) trained with Double-Sampling (DS) and Inaccurate Parameter Models (IPM) uncertainty. The results presented in Table 1 consistently demonstrate that `GRAD` achieves competitive or superior performance compared to baseline methods in each perturbed environment, showcasing its effectiveness in robustly handling transition uncertainty.

**Short-term Memorized Temporall-coupled Attacks.** While our temporally-coupled setting considering perturbation from the last time step aligns with the common practice of state adversaries, which typically perturb the current state without explicitly attacking short-term memory, we recognized the importance of exploring a more general scenario akin to a general partially observable MDP (Efroni et al., 2022). We introduced a short-term memorized temporally-coupled attacker by calculating the mean of perturbations from the past 10 steps and applying the temporally-coupled constraint to this mean. The results in Table 2 from these additional experiments against short-term memorized temporally-coupled attacks underscore the efficacy of GRAD under this extended setting. GRAD consistently demonstrates heightened robustness compared to other robust baselines when confronted with a memorized temporally-coupled adversary. These findings provide valuable insights into the

| Perturbed Environments | | RNAC-PPO (DS) | RNAC-PPO (IPM) | **GRAD** |
|---|---|---|---|---|
| Hopper | Natural reward | $3502 \pm 256$ | $3254 \pm 138$ | $3482 \pm 209$ |
| | 'leg_joint_stiffness' | $2359 \pm 182$ | $2289 \pm 124$ | $2692 \pm 236$ |
| Walker | Natural reward | $4322 \pm 289$ | $4248 \pm 89$ | $4359 \pm 141$ |
| | 'foot_joint_stiffness' | $4078 \pm 297$ | $4129 \pm 78$ | $4204 \pm 132$ |
| Halfcheetah | Natural reward | $5524 \pm 178$ | $5569 \pm 232$ | $6047 \pm 241$ |
| | 'back_actuator_range' | $768 \pm 102$ | $1143 \pm 45$ | $1369 \pm 117$ |

**Table 1:** Comparison of cumulative reward in Perturbed Environments with changed physical parameters.

| Short-term Memorized Temporally-Coupled Attacks | Hopper | Walker2d | Halfcheetah | Ant | Humanoid |
|---|---|---|---|---|---|
| PA-ATLA-PPO | $2334 \pm 249$ | $2137 \pm 258$ | $3669 \pm 312$ | $2689 \pm 189$ | $1573 \pm 232$ |
| WocaR-PPO | $2256 \pm 332$ | $2619 \pm 198$ | $4228 \pm 283$ | $3229 \pm 178$ | $2017 \pm 213$ |
| **GRAD** | $\mathbf{2869 \pm 228}$ | $\mathbf{3134 \pm 251}$ | $\mathbf{4439 \pm 287}$ | $\mathbf{3617 \pm 188}$ | $\mathbf{2736 \pm 269}$ |

**Table 2:** Performance Comparison under Memorized Temporally-Coupled Attacks

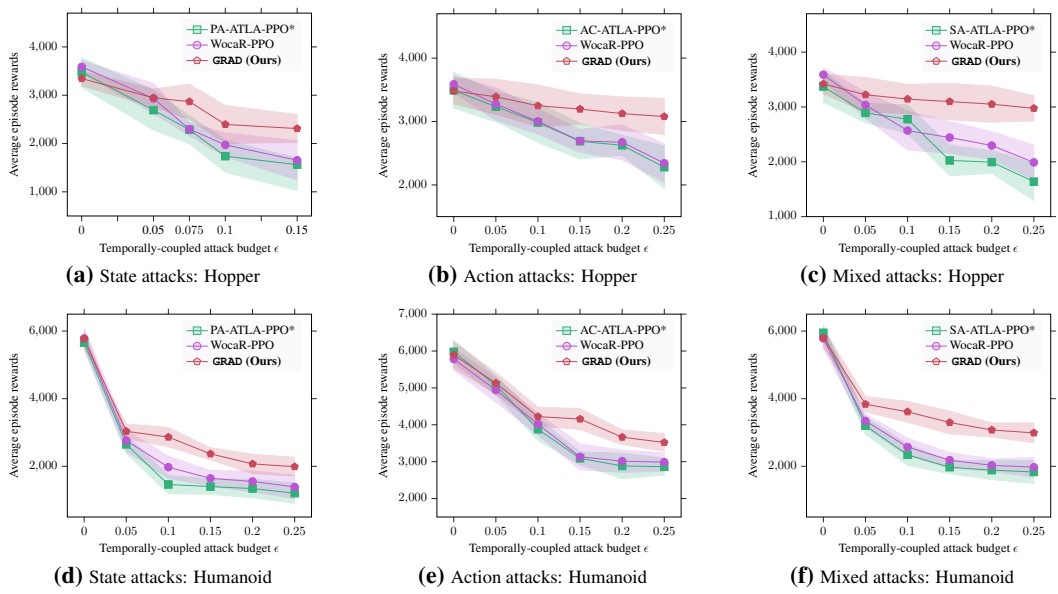

**(a)** State attacks: Hopper  **(b)** Action attacks: Hopper  **(c)** Mixed attacks: Hopper

**(d)** State attacks: Humanoid  **(e)** Action attacks: Humanoid  **(f)** Mixed attacks: Humanoid

**Figure 9:** Comparisons under state or action or mixed temporally-coupled attacks w.r.t. diverse attack budgets $\epsilon$'s on Hopper and Humanoid.

temporal scope of perturbations, contributing to a more comprehensive understanding of GRAD's capabilities in handling diverse adversarial scenarios.

## C.3 ATTACK BUDGETS

In Figure 9, we report the performance of baselines and GRAD under different attack budget $\epsilon$. As the value of $\epsilon$ increases, the rewards of robust agents under different types of attacks decrease accordingly. However, our approach consistently demonstrates superior robustness as the attack budget changes.

## C.4 TEMPORALLY-COUPLED CONSTRAINTS

We also investigate the impact of temporally-coupled constraints $\bar{\epsilon}$ on attack performance, as we explained in our experiment section.

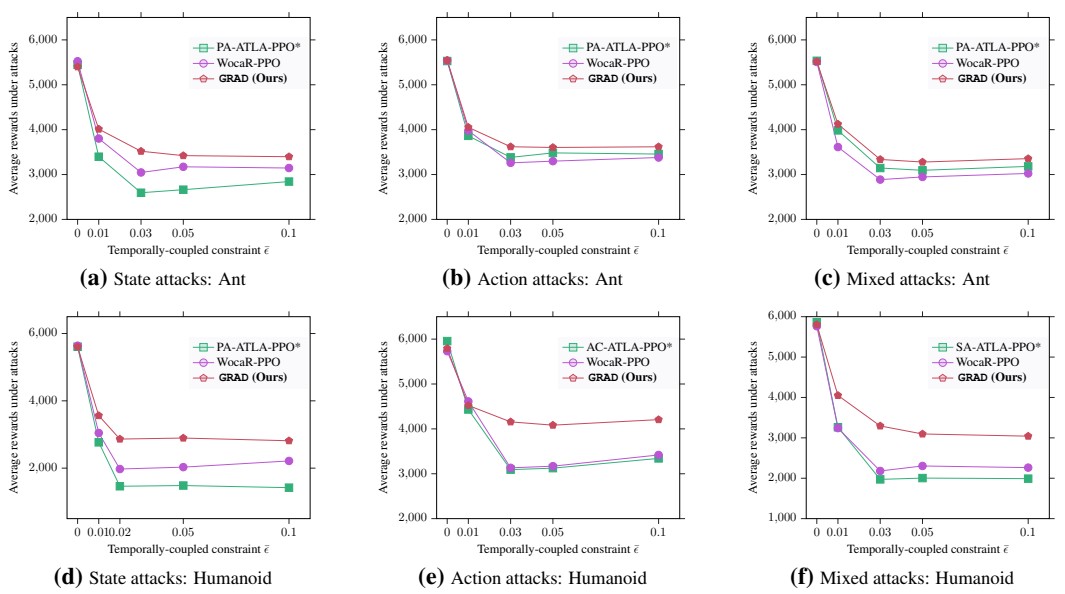

**Figure 10:** Comparisons under state or action or mixed temporally-coupled attacks with diverse temporally-coupled constraints $\epsilon$'s on Ant and Humanoid.

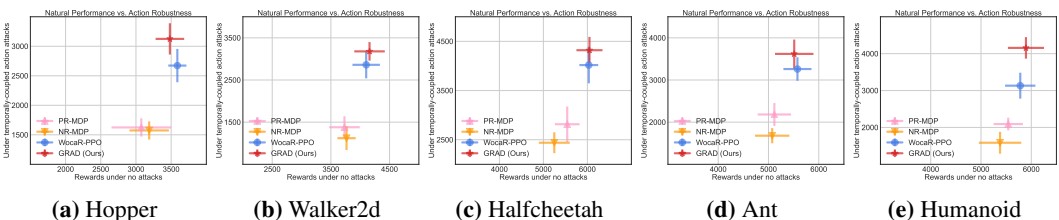

**Figure 11:** Natural performance vs. Robustness against temporally-coupled action perturbations

### C.5 NATURAL REWARD VS. ROBUSTNESS

We presents the natural performance comparison of GRAD and action robust baselines in Figure 11.

### C.6 COMPUTATIONAL COST

The training time for GRAD can vary depending on the specific environment and its associated difficulty. Typically, on a single V100 GPU, training GRAD takes around 20 hours for environments like Hopper, Walker2d, and Halfcheetah. However, for more complex environments like Ant and Humanoid, the training duration extends to approximately 40 hours. It's worth noting that the training time required for defense against state adversaries or action adversaries is relatively similar.

