# OpenReview forum: "Game-Theoretic Robust Reinforcement Learning Handles Temporally-Coupled Perturbations"
_ICLR.cc/2024/Conference — ICLR 2024 poster_

### Official Review · Reviewer_GZCq · 2023-10-23

**Soundness:** 2 fair
**Presentation:** 2 fair
**Contribution:** 2 fair
**Rating:** 5
**Confidence:** 4

**Summary:**

This paper addresses the challenge of deploying reinforcement learning (RL) systems that can withstand uncertainties, particularly those that are temporally coupled. Recognizing that conventional robust RL methods may falter against such temporally-linked perturbations, the authors introduce a game-theoretic approach called GRAD (Game-theoretic Response approach for Adversarial Defense). GRAD conceptualizes the robust RL problem as a partially observable two-player zero-sum game and uses Policy Space Response Oracles (PSRO) to achieve adaptive robustness against evolving adversarial strategies. The study's experiments confirm GRAD's performance in ensuring RL robustness against both temporally coupled and standard adversarial perturbations.

**Strengths:**

The authors formulate the robust RL objective as a zero-sum games and demonstrating the efficacy of game-theoretic RL in tackling this objective.

**Weaknesses:**

1. This paper does not have a clear mathematical representation of the problem it intends to address.

2. The article claims its primary contribution lies in using zero-sum games to formulate the robust RL problem. However, employing zero-sum games to account for uncertainties, whether in single-agent or multi-agent RL, is well-established, as seen in works like Robust Adversarial Reinforcement Learning, Robust Reinforcement Learning as a Stackelberg Game via Adaptively-Regularized Adversarial Training, and Robust Multi-Agent Reinforcement Learning with State Uncertainty. Given this widespread application, the paper's stated novelty becomes questionable.

3. In terms of algorithmic design, the proposed method is largely an application of the Policy-Space Response Oracles (PSRO). The novelty seems limited, and it's unclear how PSRO uniquely addresses the issue of temporally-coupled perturbations.

4. Considering that the PSRO algorithm converges to an NE in two-player, zero-sum games and has seen recent extensions to other equilibria types [1, 2], the paper's proposed method, essentially a reiteration of PSRO, makes the convergence proof for GRAD appear somewhat lackluster in its contribution.

5. This paper has ample room for improvement in writing, problem formulation, and the method itself. For instance, by simply using the triangle inequality, we could give the range of $\bar{\epsilon}$ that break the temporally-coupled property, rather than merely stating, "By setting \epsilon ̄ to a large value, it converges to the non-coupled attack scenario." Additionally, the motivation behind temporally-coupled perturbations lacks clarity and persuasiveness, leaving me unconvinced of its pressing relevance.

[1] Lanctot, Marc, et al. "A unified game-theoretic approach to multiagent reinforcement learning." Advances in neural information processing systems 30 (2017).

[2] McAleer, Stephen, et al. "Pipeline psro: A scalable approach for finding approximate nash equilibria in large games." Advances in neural information processing systems 33 (2020): 20238-20248.

**Questions:**

Please see the Weaknesses, I will decide the final rating after the rebuttal.

**Details Of Ethics Concerns:**

Non.

---

> ### Author Response · Authors · 2023-11-15
> **Response to Reviewer GZCq's concern about problem representation and contributions**
>
> We thank Reviewer GZCq for the valuable feedback on our paper. We have provided detailed responses to the concerns raised by the reviewer and updated our paper, incorporating clarifications and discussions where necessary.
>
> > 1. This paper does not have a clear mathematical representation of the problem it intends to address.
>
> We appreciate the reviewer's feedback regarding the clarity of our problem formulation. In response to this concern, we have enhanced the paper by providing a more explicit and detailed formulation of our problem in Section 2 based on the zero-sum game. We thank the reviewer for highlighting the mathematical problem formulation.
>
> > 2. The article claims its primary contribution lies in using zero-sum games to formulate the robust RL problem. However, employing zero-sum games to account for uncertainties, whether in single-agent or multi-agent RL, is well-established. Given this widespread application, the paper's stated novelty becomes questionable.
>
> While we acknowledge the precedent exploration of zero-sum games in robust RL within existing works, it is crucial to emphasize the distinctive contribution of our method, GRAD.
>
> * **Converge to approximate equilibriums on an adversary population.**
>
> During training, our approach achieves an approximate equilibrium on an adversary policy set, which is totally different from other robust RL papers which use the zero-sum game formulation but only learn with an adversary during alternating training. GRAD has the capability to continuously explore and learn new policies as best responses that are not present in the current policy set, which allows for a more thorough exploration of the policy space.
>
> * **Adaptable to different adversaries.**
>
> Significantly, prior works, such as [1, 2, 3], primarily focus on a specific adversary or only consider the worst-case scenario, often adopting a highly pessimistic outlook that limits their adaptability to adversaries with different constraints. However, GRAD does not specifically target certain adversaries during training. Instead, at each epoch, it learns the best response to the adversary policy sampled from the adversary population, **making it adaptable to different adversarial scenarios, including both temporally-coupled and non-temporally-coupled adversaries**.
>
> Furthermore, previous approaches exclusively consider specific attack domains for adversaries, such as perturbations on environmental dynamics [1, 2] and action perturbations [3]. In contrast, GRAD represents a pioneering effort by introducing the first robust RL framework that accommodates various adversaries including action, state, and transition adversaries, **without any constraints on attack domains**. This broader scope positions GRAD as a more practical and scalable general solution for robust RL.
> * **Solving adversarial RL rather than uncertainty-aware RL.**
>
> Our method is proposed to defend against adversarial perturbations and improve adversarial robustness, while many papers such as [4] only focus on uncertainty-aware RL. Though GRAD exhibits strong performance even under model uncertainty, as shown in our experiments. Adversarial RL where the learning process is intentionally disrupted by an adaptive adversary (as a maxmin problem) is more challenging than uncertainty-aware RL (a max problem under uncertainty).
>
> To provide further clarity on this aspect, we have included a detailed illustration of robust RL formulated as a zero-sum game in **Appendix B (Additional Related Work)** and improved our writing in sections of introduction and methodology. We believe these additions effectively address your concerns.
>
> [1] Robust Adversarial Reinforcement Learning, Pinto et al. 2017.
>
> [2] Robust Reinforcement Learning as a Stackelberg Game via Adaptively-Regularized Adversarial Training, HUang et al. 2021.
>
> [3] Action Robust Reinforcement Learning and Applications in Continuous Control, Tessler et al. 2019.

---

> > ### Author Response · Authors · 2023-11-15
> > **Response to Reviewer GZCq's concern about novelty and convergence proof**
> >
> > > 3. The proposed method is largely an application of the Policy-Space Response Oracles (PSRO). The novelty seems limited, and it's unclear how PSRO uniquely addresses the issue of temporally-coupled perturbations.
> >
> > The question prompts us to articulate our novelty concerning the PSRO framework more explicitly. Although GRAD draws inspiration from the PSRO framework, it innovatively extends and adapts the original concept to effectively tackle robust RL challenges. The novelty of GRAD lies in its scalability and adaptability in robust RL against diverse types of adversaries. **Our primary contribution centers on demonstrating that game-theoretic RL can be a general and scalable solution for the robust RL objective**, transcending the confines of a particular game-theoretic RL algorithm like PSRO. In our revised paper, we have enriched the introduction and method sections with additional illustrations to provide a clearer depiction of our contributions.
> >
> > * **How GRAD uniquely addresses temporally-coupled perturbations**
> >
> > GRAD introduces attackers under temporally-coupled constraints during training, enhancing efficiency, and resulting in a robust model against temporally-coupled and non-temporally-coupledadversaries. GRAD's unique ability to address diverse adversaries from its achievement of the approximate equilibrium after training with a varied adversary policy set. Notably, while GRAD is not exclusively designed for temporally-coupled attacks, its effectiveness in this specific and meaningful case highlights the contribution of our framework.
> >
> >
> > > 4. Considering that the PSRO algorithm converges to an NE in two-player, zero-sum games and has seen recent extensions to other equilibria types [1, 2], the paper's proposed method, essentially a reiteration of PSRO, makes the convergence proof for GRAD appear somewhat lackluster in its contribution.
> >
> > We appreciate the reviewer's attention to the convergence proof aspect of our proposed method, GRAD. While it is true that PSRO has seen various extensions, our primary contribution lies in pioneering the application of PSRO to the robust RL setting. This adaptation is a novel and significant step in leveraging the power of PSRO to enhance the adversarial robustness. While theoretical convergence provides valuable insights, our emphasis is on the practical contribution of GRAD in achieving robustness against diverse adversarial attacks. The convergence proof serves as a foundational aspect supporting the effectiveness of our method.

---

> > > ### Author Response · Authors · 2023-11-15
> > > **Response to Reviewer GZCq's concern about writing and motivation**
> > >
> > > > 5. Writing improvement and motivation behind temporally-coupled perturbations
> > >
> > > We express our sincere gratitude for the valuable feedback provided by the reviewer, and we have worked to address the writing improvements and revised our introduction, preliminary, and methodology sections. In response to the reviewer's concerns, we offer the following clarifications for the motivation behind temporally-coupled perturbations:
> > >
> > > * **Practical realism**: We emphasize that the introduction of temporally-coupled constraints serves to create a more practical and realistic setting for adversarial perturbations, aligning with real-world scenarios where adversaries may exhibit temporal correlations.
> > > For example, the occurrence of abrupt forces on robots often comes from the same direction or disturbance source over a period of time. This necessitates our consideration of the temporal correlation of perturbations.
> > >
> > > * **Efficient search space**: Temporally-coupled constraints significantly reduce the search space for potential attackers. This reduction enhances the efficiency of the exploration process, contributing to the scalability and applicability of our approach.
> > >
> > > * **Novel Case to challenge existing methods**: Temporally-coupled perturbations represent a novel case to challenge existing methods that only consider l-p attacks. Our empirical experiments demonstrate that, even with the introduction of temporally-coupled constraints, these perturbations can have a notable impact on existing robust models, showcasing the need for considering such scenarios in robust RL.
> > >
> > > We sincerely hope that these clarifications and paper improvements align with the reviewer's expectations. We welcome further discussions and suggestions to enhance the quality of our paper.

---

> > > > ### Author Response · Authors · 2023-11-20
> > > > **Does our response address your concerns?**
> > > >
> > > > Dear reviewer GZCq,
> > > >
> > > > As the review discussion stage is drawing to a close, we sincerely hope that you will take the time to review both our revised paper and our response. We genuinely believe that our response have addressed all raised concerns and look forward to any additional feedback. If there are any further questions or points of clarification needed, please do not hesitate to inform us. We appreciate the opportunity for continued discussion and improvement.

---

> > > > > ### Comment · Reviewer_GZCq · 2023-11-20
> > > > > **Thanks for the response**
> > > > >
> > > > > Dear authors,
> > > > >
> > > > > Thanks for the response and all the efforts to resolve my concerns. I have raised my score since the paper writing has been improved.

---

> > > > > > ### Author Response · Authors · 2023-11-22
> > > > > > **Kindly request any additional feedback for the last day**
> > > > > >
> > > > > > We greatly appreciate your recent review and the adjustment of the score to 5. Your insights have been instrumental in shaping the quality of our paper.
> > > > > >
> > > > > > As today marks the final day of the discussion period, we are curious to understand any specific aspects that still place our paper below the borderline for acceptance. Your perspectives are invaluable to us, and we are eager to ensure that we address any remaining issues to meet your expectations.
> > > > > >
> > > > > > Thank you so much for your time and thoughtful consideration. We look forward to receiving any further suggestions you may have before the conclusion of the discussion period.

---

> ### Author Response · Authors · 2023-11-21
> **Response to Reviewer GZCq**
>
> Dear Reviewer GZCq,
>
> Thank you so much for your prompt response and for improving our scores. We sincerely appreciate the valuable insights you provided for our paper, which have significantly contributed to enhancing its quality. If there are any further questions or concerns, we would be more than happy to address them.

---

### Official Review · Reviewer_JmX3 · 2023-10-27

**Soundness:** 4 excellent
**Presentation:** 4 excellent
**Contribution:** 3 good
**Rating:** 6
**Confidence:** 5

**Summary:**

This work introduces a novel-framework of temporally-coupled robust RL problem that is closer to the real-world setting. This work proposes GRAD, a game-theoretic approach to provide robust policies played against an adversary which attacks states and actions fitting in the temporally-coupled robust RL problem setting. This work also gives extensive complementing experiment results.

**Strengths:**

I think this work really pushes the robust RL community research efforts further by answering:

> can we design robust RL algorithms for realistic nature attacks?

The main contribution of game-theoretic algorithm with temporal-based nature attacks (robust RL problem) is a really nice idea worthy for publication. But the score reflects my weakness section.

**Weaknesses:**

I have only a few weakness for this work as follows:

> The current framework considers robustness against state and action uncertainty. More closer work [1] and thereafter are not included. Model uncertainty is justified in the framework mentioning the evolution of the environment depends on the perturbed actions. Model uncertainty in robust RL is defined in more generality [2-10]. So it will be better to include more detailed Related Works including [2-10] and more relevant works in the revision. I agree this work includes experiments with model uncertainty, but the baselines are also only action robust algorithms. I'd rather see more extensive writing and experiments for model-uncertainty OR the current work just focusing on state-action uncertainty is a big step forwards in itself. I've also stopped at '10' since you get the idea of inadequate related work discussion.

> GRAD shares similar idea of RARL algorithm (Pinto et al., 2017), that is, zero-sum structure to get the robust policy against the nature adversary. More details than below must be added to point out key differences (like state-action uncertainty inbuilt) and due references need to be given.
` Pinto et al. (Pinto et al., 2017) model the competition between the agent and the attacker as a zero-sum two-player game, and train the agent under a learned attacker to tolerate both environment shifts and adversarial disturbances `

I am open to discussions with the authors and reviewers to increase/maintain (already reflects the positive impact) my score. All the best for future decisions!

[1] Robust Multi-Agent Reinforcement Learning with State Uncertainty Sihong He, Songyang Han, Sanbao Su, Shuo Han, Shaofeng Zou, and Fei Miao. Transactions on Machine Learning Research, June 2023.

[2] Xu. Z, Panaganti. K, Kalathil. D, Improved Sample Complexity Bounds for Distributionally Robust Reinforcement Learning. Artificial Intelligence and Statistics, 2023.

[3] Nilim, A. and El Ghaoui, L. (2005). Robust control of Markov decision processes with uncertain transition matrices. Operations Research, 53(5):780–798

[4] Iyengar, G. N. (2005). Robust dynamic programming. Mathematics of Operations Research, 30(2):257–280.

[5] Panaganti, K. and Kalathil, D. (2021). Robust reinforcement learning using least squares policy iteration with provable performance guarantees. In Proceedings of the 38th International Conference on Machine Learning, pages 511–520.

[6] Panaganti, K. and Kalathil, D. (2022). Sample complexity of robust reinforcement learning with a generative model. In Proceedings of The 25th International Conference on Artificial Intelligence and Statistics, pages 9582–9602.

[7] Roy, A., Xu, H., and Pokutta, S. (2017). Reinforcement learning under model mismatch. In Advances in Neural Information Processing Systems, pages 3043–3052.

[8] Panaganti, K., Xu, Z., Kalathil, D., and Ghavamzadeh, M. (2022). Robust reinforcement learning using offline data. Advances in Neural Information Processing Systems (NeurIPS).

[9] Shi, L. and Chi, Y. (2022). Distributionally robust model-based offline reinforcement learning with near-optimal sample complexity. arXiv preprint arXiv:2208.05767

[10] L Shi, G Li, Y Wei, Y Chen, M Geist, Y Chi  (2023) The Curious Price of Distributional Robustness in Reinforcement Learning with a Generative Model, NeurIPS 2023

**Questions:**

-n/a-

---

> ### Author Response · Authors · 2023-11-15
> **Response to Reviewer JmX3's concern about related discussions for model uncertainty**
>
> Thanks Reviewer JmX3 for acknowledging and encouraging our paper's contributions. We provide a detailed response to your insightful feedback and make detailed revisions to our paper accordingly.
>
> > 1. The current framework considers robustness against state and action uncertainty. It will be better to include more detailed Related Works including and more relevant works in the revision.
>
> We appreciate your valuable suggestion to provide a more detailed discussion on related works concerning robustness against uncertainty. Following your advice, we have expanded the discussion in **Appendix B to include the mentioned papers**, aiming to make our literature survey more comprehensive.
>
> Regarding the experiments on state robustness, we present results under the most robust state adversary. Other state attack methods or state uncertainty sets are less effective in attacking both robust baselines and our approach. Furthermore, we need to clarify that our method is proposed to **defend against adversarial perturbations and improve adversarial robustness**, which differs from many papers that focus on uncertainty-aware RL. Adversarial RL where the learning process is intentionally disrupted by an adaptive adversary (as a max-min problem) is more challenging than uncertainty-aware RL (a max problem under uncertainty)many papers focus on uncertainty-aware RL. Remarkably, GRAD exhibits strong performance even under model uncertainty, as shown in our experiments.
>
> We have included **an additional set of experiments and updated results in Appendix C**. These experiments consider model mismatch or transition uncertainty following [1], across three perturbed MuJoCo environments with changes in different physical parameters. The observed performance disparity underscores the effectiveness of GRAD and its robustness against model uncertainty.
>
> |  Perturbed Environments  |                    | RNAC-PPO(DS) | RNAC-PPO(IPM) | **GRAD** |
> |----------------|----------------|------------|------------|-----------------|
> | Hopper | natural reward | **3502 ± 256**       | 3254 ± 138       |  3482 ± 209   |
> |                                   | 'leg_joint_stiffness' value: 30   | 2359 ± 182       | 2289  ± 124       |  **2692  ± 236**     |
> | Walker | natural reward | 4322 ± 289        | 4248 ± 89       |  **4359 ± 141**   |
> |                                   | 'foot_joint_stiffness' value: 30    | 4078 ± 297       |  4129 ± 78       |   **4204 ± 132**     |
> | Halfcheetah | natural reward | 5524 ± 178        | 5569 ± 232      |  **6047 ± 241**   |
> |                                   | Bound on 'back_actuator_range': 0.5   | 768 ± 102       |  1143 ± 45       |   **1369 ± 117**     |
>
> [1] Natural Actor-Critic for Robust Reinforcement Learning with Function Approximation, Zhou et al. 2023.

---

> ### Author Response · Authors · 2023-11-15
> **Response to Reviewer JmX3's concern about novelty of zero-sum game formulation**
>
> > 2. GRAD shares similar idea of RARL algorithm (Pinto et al., 2017), that is, zero-sum structure to get the robust policy against the nature adversary.
>
> This is indeed a valuable feedback. In Appendix B, we have extensively discussed other robust RL works that utilize zero-sum games and highlighted the distinctions between GRAD and these approaches.
>
> * **Converge to approximate equilibriums on an adversary population**
>
> During training, our approach achieves an approximate equilibrium on an adversary policy set, which is totally different from other robust RL papers which use the zero-sum game formulation but only learn with an adversary during alternating training. GRAD has the capability to continuously explore and learn new policies as best responses that are not present in the current policy set, which allows for a more thorough exploration of the policy space.
>
> * **Adaptable to different adversaries**
>
> Significantly, prior works, such as [2, 3, 4], primarily focus on a specific adversary or only consider the worst-case scenario, often adopting a highly pessimistic outlook that limits their adaptability to adversaries with different constraints. However, GRAD does not specifically target certain adversaries during training. Instead, at each epoch, it learns the best response to the adversary policy sampled from the adversary population, **making it adaptable to different adversarial scenarios, including both temporally-coupled and non-temporally-coupled adversaries**.
>
> Furthermore, previous approaches exclusively consider specific attack domains for adversaries, such as perturbations on environmental dynamics [2, 3] and action perturbations [4]. In contrast, GRAD represents a pioneering effort by introducing the first robust RL framework that accommodates various adversaries including action, state and transition adversaries, **without any constraints on attack domains**. This broader scope positions GRAD as a more practical and scalable general solution for robust RL.
>
>
> In summary, we appreciate the reviewer's identification of weaknesses in our related works discussion and our novelty in problem formulation, while also acknowledging our contributions. We look forward to further discussions.
>
>
> [2] Robust Adversarial Reinforcement Learning, Pinto et al. 2017.
>
> [3] Robust Reinforcement Learning as a Stackelberg Game via Adaptively-Regularized Adversarial Training, HUang et al. 2021.
>
> [4] Action Robust Reinforcement Learning and Applications in Continuous Control, Tessler et al. 2019.

---

> > ### Author Response · Authors · 2023-11-21
> > **Does our response address your concerns?**
> >
> > Dear Reviewer JmX3,
> >
> > Thank you once again for your positive review of our paper, which is highly encouraging as we strive to enhance the quality of our work. We appreciate your concerns regarding novelty and related works, and we have provided detailed explanations to address them. As the review discussion stage is concluding, we hope you'll take the time to review both our revised paper and our response. If there are further questions, additional feedback, or more clarifications needed, such as additional aspects of related works that you find essential for discussion, please do not hesitate to inform us. We are dedicated to making any necessary revisions to ensure your satisfaction.

---

> ### Comment · Reviewer_JmX3 · 2023-11-21
> **Ack**
>
> Thank you for the detailed response. I do think the authors have improved the manuscript compared to the pre-rebuttal stage and also seem to have addressed the major concerns of all reviewers. I will update my score after authors-reviewers discussions.

---

> > ### Author Response · Authors · 2023-11-21
> > **Response to Reviewer JmX3**
> >
> > Thank you very much for your prompt response and appreciation of our paper. We highly value your insightful review comments! We will further incorporate the insights from the author-reviewer discussion period into our global response and our paper. We appreciate the time and effort you dedicated during the review and discussion stage.

---

### Official Review · Reviewer_mjop · 2023-10-30

**Soundness:** 2 fair
**Presentation:** 2 fair
**Contribution:** 2 fair
**Rating:** 5
**Confidence:** 4

**Summary:**

The classic robust RL focuses on worst-case scenarios, which may result in an overly conservative policy. Instead, this paper introduces temporally-coupled perturbations. Additionally, this paper proposed an adversarial training approach named the game-theoretic response approach for adversarial defense. Finally, the authors show the robust performance of the proposed methods in several MuJoCo tasks compared with several baselines.

**Strengths:**

* This paper is easy to follow.
* This paper does thorough experiments for state attacks and action attacks and compares the proposed method with several baselines.
* The temporally-coupled adversarial perturbation seems new.

**Weaknesses:**

* Although the temporally-coupled adversarial perturbation seems new, it is quite limited. Definition 3.2 only considers the temporally-coupled perturbation from the last time step. Even if the authors don't consider the general partially observable MDP, they should consider a more general case, e.g., m-order MDP [Efroni et al. 2022, Provable Reinforcement Learning with a short-term memory].
* The zero-sum game-based approach is not new for robust training in RL, e.g., [Tessler et al., 2019].
* This paper misses one classic setting of robust MDP (i.e., transition adversaries) [e.g., Iyengar'05, Robust Dynamic Programming] as well as related baselines [e.g., Zhou et al. 2023, Natural Actor-Critic for Robust Reinforcement Learning with Function Approximation].
* Figure 1 doesn't make sense to me. The robust baseline considers the worst-case scenario, which should be more stable for different kinds of attacks compared with the less conservative model that is proposed in this work.

**Questions:**

Please refer to the "weakness" section for further information.

---

> ### Author Response · Authors · 2023-11-15
> **Response to Reviewer mjop's concern about attack settings and novelty of zero-sum game formulation**
>
> Thanks Reviewer mjop for the insightful review of our paper. We acknowledge the concern the reviewer raised. We have updated our paper and provided a comprehensive response to address the issues highlighted.
>
> > 1. Limitation of our temporally-coupled settings. Definition 3.2 only considers the temporally-coupled perturbation from the last time step. Even if the authors don't consider the general partially observable MDP, they should consider a more general case, e.g., m-order MDP [Efroni et al. 2022, Provable Reinforcement Learning with a short-term memory].
>
> According to the reviewer's suggestion, we have expanded our experiments to include a more general scenario and introduce a memorized temporally-coupled attack. We calculate the mean of perturbations from the 10 past steps and apply the temporally-coupled constraint on this past perturbation mean. We add this experiment in **Appendix C.2** to showcase the effectiveness of GRAD under this extended adversarial setting. The added results indicate that **GRAD exhibits enhanced robustness compared to other robust baselines under the memorized temporally-coupled attack**. We believe this addition contributes valuable insights to the paper and addresses your concerns regarding the temporal scope of perturbations.
>
> Moreover, the primary reason we focused on perturbations from the last time step is grounded in the common practice of state adversaries, which typically perturb the current state without explicitly attacking short-term memory, but we agree that considering the short-term memory is a reasonable and considerable setting for adversaries. We eagerly anticipate any additional discussions or suggestions the review may have.
>
> | Memorized temporally-coupled Attacks| Hopper | Walker2d | Halfcheetah | Ant | Humanoid |
> | -------- | -------- | -------- |-------- | -------- |-------- |
> | PA-ATLA-PPO     | 2334 ± 249     | 2137 ± 258    | 3669 ± 312    | 2689 ± 189     | 1573 ± 232
> | WocaR-PPO     | 2256 ± 332     | 2619 ± 198    | 4228 ± 283     | 3229 ± 178     | 2017 ± 213
> | **GRAD** |  **2869 ± 228**     | **3134 ± 251**    | **4439 ± 287**    | **3617 ± 188** | **2736 ± 269**
>
>
> > 2. The zero-sum game-based approach is not new for robust training in RL, e.g., [Tessler et al., 2019].
>
> While we acknowledge the precedent exploration of zero-sum games in robust RL within the existing literature, it is crucial to emphasize the distinctive contribution of our method, GRAD.
>
> * **Converge to approximate equilibriums on an adversary population.**
>
> During training, our approach achieves an approximate equilibrium on an adversary policy set, which is totally different from other robust RL papers which use the zero-sum game formulation but only learn with an adversary during alternating training. GRAD has the capability to continuously explore and learn new policies as best responses that are not present in the current policy set, which allows for a more thorough exploration of the policy space.
>
> * **Adaptable to different adversaries.**
>
> Significantly, prior works, such as [1, 2, 3], primarily focus on a specific adversary or only consider the worst-case scenario, often adopting a highly pessimistic outlook that limits their adaptability to adversaries with different constraints. However, GRAD does not specifically target certain adversaries during training. Instead, at each epoch, it learns the best response to the adversary policy sampled from the adversary population, **making it adaptable to different adversarial scenarios, including both temporally-coupled and non-temporally-coupled adversaries**.
>
> Furthermore, previous approaches exclusively consider specific attack domains for adversaries, such as perturbations on environmental dynamics [1, 2] and action perturbations [3]. In contrast, GRAD represents a pioneering effort by introducing the first robust RL framework that accommodates various adversaries including action, state, and transition adversaries, **without any constraints on attack domains**. This broader scope positions GRAD as a more practical and scalable general solution for robust RL.
>
> To provide further clarity on this aspect, we have included a detailed illustration of robust RL formulated as a zero-sum game in **Appendix B (Additional Related Work)** and improved our writing in sections of introduction and methodology. We believe these additions effectively address your concerns.
>
> [1] Robust Adversarial Reinforcement Learning, Pinto et al. 2017.
>
> [2] Robust Reinforcement Learning as a Stackelberg Game via Adaptively-Regularized Adversarial Training, HUang et al. 2021.
>
> [3] Action Robust Reinforcement Learning and Applications in Continuous Control, Tessler et al. 2019.

---

> ### Author Response · Authors · 2023-11-15
> **Response to Reviewer mjop's concern about experiments for transition adversaries and Figure 1's illustration**
>
> > 3. This paper misses one classic setting of robust MDP (i.e., transition adversaries) [e.g., Iyengar'05, Robust Dynamic Programming] as well as related baselines [e.g., Zhou et al. 2023, Natural Actor-Critic for Robust Reinforcement Learning with Function Approximation].
>
> We deeply appreciate the insightful suggestion regarding the incorporation of experiments involving transition adversaries. To attempt to study this great point, we conducted a meticulous comparative study pitting GRAD against the baseline RNAC-PPO trained with DS and IPM uncertainty sets across three perturbed MuJoCo environments with changed physic parameters, following the experimental protocols outlined in [4].
>
> The results unequivocally demonstrate that **GRAD outperforms RNAC-PPO across all tasks under transition uncertainty**. This performance disparity underscores the efficacy of GRAD and its adaptability across diverse attack domains. We believe these results findings make substantial empirical contributions to the paper, effectively addressing the concern raised by the reviewer. We also add this experiment in **Appendix C.2**.
>
> |  Perturbed Environments  |                    | RNAC-PPO(DS) | RNAC-PPO(IPM) | **GRAD** |
> |----------------|----------------|------------|------------|-----------------|
> | Hopper | natural reward | **3502 ± 256**       | 3254 ± 138       |  3482 ± 209   |
> |                                   | 'leg_joint_stiffness' value: 30   | 2359 ± 182       | 2289  ± 124       |  **2692  ± 236**     |
> | Walker | natural reward | 4322 ± 289        | 4248 ± 89       |  **4359 ± 141**   |
> |                                   | 'foot_joint_stiffness' value: 30    | 4078 ± 297       |  4129 ± 78       |   **4204 ± 132**     |
> | Halfcheetah | natural reward | 5524 ± 178        | 5569 ± 232      |  **6047 ± 241**   |
> |                                   | Bound on 'back_actuator_range': 0.5   | 768 ± 102       |  1143 ± 45       |   **1369 ± 117**     |
>
> > 4. Figure 1 doesn't make sense. The robust baseline considers the worst-case scenario, which should be more stable for different kinds of attacks compared with the less conservative model that is proposed in this work.
>
> We have revised the caption of Figure 1 to explain why the baseline is not robust to the temporally-coupled adversaries,
>
> While it is true that WocaR-PPO aims to enhance its robustness in the worst-case scenarios or improve its lower bound of performance, **it does not guarantee superiority under other case scenarios, such as temporally-coupled perturbations**. The baseline's focus on worst case estimations and improvement does not inherently ensure robustness across all conceivable situations. It is noteworthy that WocaR-PPO's relatively higher risk of stumbling under temporally-coupled attacks, as demonstrated in our experiments. This is precisely where GRAD excels. GRAD finds the equilibrium with an adversary policy set, showcasing a significant advantage in diverse adversarial settings. We believe these aspects underscore the substantial advantages of GRAD and welcome further discussion on this matter.
>
> In summary, we have responded to the reviewer's concern by providing additional experiments in temporally-coupled settings and diverse attack domains, illustrating GRAD's robustness. We also discuss our noverlty different from related works to better elucidate GRAD's strengths. Regarding Figure 1, we explained the observations and modified Figure 1's illustration in our paper. We appreciate the valuable feedback, which enhances the quality of our paper, and we look forward to further discussion and additional suggestions.
>
> [4] Natural Actor-Critic for Robust Reinforcement Learning with Function Approximation, Zhou et al. 2023.

---

> ### Comment · Reviewer_mjop · 2023-11-19
>
> Based on the author's efforts to solve some concerns, I have increased the score. Additionally, I don't completely agree with the authors' claim that "game-theoretic RL can be a general and scalable solution for the robust RL objective". Not only does the game-theoretic method bring instability, but it also makes training take much longer compared with the standard non-robust approach. Instead, there exist some computationally efficient methods to guarantee robustness theoretically and empirically (e.g., RNAC [4]). In other words, I am wondering if it is necessary to make slight improvements for robustness at the expense of significant training time.

---

> ### Author Response · Authors · 2023-11-20
> **Response to Reviewer mjop**
>
> Thank you very much for your further response and raising a discussion-worthy question. We would like to clarify the following points.
> 1. Prior methods, such as RNAC, provide theoretical and empirical guarantees for robustness, focusing on uncertainty-aware RL. In contrast, our method mainly aims to enhance robustness specifically for adversarial RL, while our experiments also demonstrate GRAD's significant robustness against model uncertainty. Adversarial RL involves a max-min problem against a learnable and adaptive adversary, posing a greater challenge than uncertainty-aware RL, which is a max problem under uncertainty. Existing methods for adversarial RL often struggle to provide both theoretical guarantees and adaptability to different adversaries, often at a higher computational cost, such as alternating training with learned adversaries. The potential superiority of game-theoretic methods in adversarial RL arises from the fact that adversaries can adjust/learn their attack strategies based on the agent's policy. Hence, it becomes necessary to find the approximate equilibrium solutions to discover more robust strategies against adversarial perturbations
>
> 2. We understand your concern, and if our method were only effective with minor improvements in robustness against a limited set of uncertainties or perturbations, the extensive training time would indeed be unjustifiable. Therefore, we conducted experiments to demonstrate the broad practicality of our method against various adversarial perturbations with different constraints or in different attack domains. We are the first to defend against such a diverse range of perturbations, contributing to adversarial robustness.
>
> 3. Additionally, addressing computational efficiency, we propose in the discussion and limitations section that future directions could explore more efficient alternatives to PSRO. Our primary contribution lies in showcasing the efficacy of game-theoretic methods in robust RL, particularly against adversarial perturbations, providing a comprehensive and effective solution.
>
> We hope our response further addresses your concerns and helps you understand the contribution of our work to the robust RL community. We welcome further discussion, and we will appropriately supplement our discussion in the Appendix.

---

> > ### Comment · Reviewer_mjop · 2023-11-21
> >
> > Thank you for your reply. I don't completely agree with the authors' claim. Robust MDP belongs to distributional robust optimization (DRO) instead of uncertainty-aware RL. Both adversarial training and DRO involve a max-min problem, and they are two different approaches to solving robust RL problems. Additionally, action perturbations can be regarded as a special case of DRO (e.g., [Tessler et al. 2019]). But anyway, due to the author's efforts to solve concerns, I have increased the score.

---

> ### Author Response · Authors · 2023-11-21
> **Response to Reviewer mjop**
>
> Dear Reviewer mjop,
>
> Thank you once again for your response. Perhaps our previous clarification led to some misunderstandings. We want to emphasize that our main goal is to address the adversarial RL problem with an adversary that is adaptive to the agent's policy. Papers like Tessler et al. (2019) on action perturbations and Zhou et al. (2023) on model mismatch uncertainty, are hard to defend against the strongest adversarial perturbations and only empirically evaluated on uncertainty sets. We include Tessler et al. (2019) as an action-robust baseline in our paper, which is not robust under RL-based action adversaries.  This vulnerability arises due to the inherent difficulty in estimating the long-term worst-case value under adaptive adversaries. Distributional robust optimization (DRO) is also challenging to apply to this challenging problem, especially against state adversaries in the high-dimensional state space. At the same time, adversarial training methods also struggle to effectively deal with model uncertainty or model mismatch problems.
>
> Existing adversarial training methods, such as alternating training, often require simultaneous training of an RL-based adversary, leading to increased computational costs. This is worthwhile in many practical settings when we have access to a high-fidelity simulator. Our approach, leveraging game-theoretic methods, demonstrates robustness against adversarial perturbations and model uncertainty, as a more effective and general solution for high-dimensional tasks. We hope this clarifies our contributions to the robust RL community. We further elaborate on the distinctions from DRO in our Appendix.
>
> While we fully understand your concerns, it remains challenging for existing methods to effectively address adversarial perturbations, especially when considering different attack constraints and simultaneous perturbations to both state and action spaces. GRAD, however, is applicable to different adversary settings, not limited to specific uncertainty sets or adversaries confined to certain constraints. This adaptability is a key strength that sets GRAD apart from other robust RL methods.
>
> We greatly appreciate your valuable suggestions and hope for more discussions and considerations of your score before the end of the discussion period.

---

> ### Author Response · Authors · 2023-11-21
> **Response to Reviewer mjop**
>
> To provide a clearer and more detailed explanation with further discussion:
>
> 1. Regarding Tessler et al., although their formulation considers the worst-case adversary by framing robust RL as a max-min problem, their algorithm updates the adversary using only one-step gradient descent without the guarantee of adversary convergence and makes it hard to estimate the worst-case value, contributing to their lack of robustness. In contrast, our algorithm, by finding the approximate equilibrium with the adversary set, achieves stronger robustness and is more adaptable for different adversaries.
>
> 2. We acknowledge the significance of prior works and understand that your question mainly revolves around the trade-off between robustness and computational cost. Our insights center on finding the approximate equilibrium to discover optimal robust models with higher levels of model exploitation. In many practical scenarios like self-driving or robotics, where robust models are crucial, robustness under various perturbations holds more significance than computational efficiency. This is where game-theoretic methods become valuable. In game-theoretic methods, it can also balance robustness and computational efficiency by controlling the level of exploitability.
>
> We appreciate your continued engagement, and we believe this discussion has further clarified the value of our work. Looking forward to your response!

---

> > ### Comment · Reviewer_mjop · 2023-11-21
> >
> > Thank you for your further clarification. I will consider increasing the score further during the reviewer-AC discussion.

---

> ### Author Response · Authors · 2023-11-22
> **Response to Reviewer mjop**
>
> Thank you for your continued engagement and thoughtful consideration of our paper. Additional clarifications and experiments significantly contribute to the improvement of our work. Your willingness to reconsider the score for our paper is highly valued, and any further feedback you may have is welcomed.
>
> Additionally, we will express our gratitude in the comments provided to the AC for your diligent review and discussion. We truly appreciate the time, effort, and professional insights you've invested in our discussions, which are crucial for enhancing the quality of our work.

---

### Author Response · Authors · 2023-11-15
**Global response to all reviewers**

## Summary of Review and Highlights:
We express our gratitude to all the reviewers for their valuable feedback, helpful suggestions, and insightful questions. We appreciate the recognition from Reviewers mjop and JmX3 regarding our proposal of temporally-coupled attacks, as well as the acknowledgment from Reviewers GZCq and mjop of our empirical contributions. We have provided comprehensive explanations to address all raised concerns and implemented necessary revisions to enhance the quality of our paper based on reviewers' feedback.

## Review Concerns
All reviewers expressed concerns about the novelty of our zero-sum game formulation.
Reviewer mjop/JmX3: Suggested experiments for additional adversaries or uncertainty.
Reviewer mjop: Proposed an extended setting for temporally-coupled perturbations.
Reviewer mjop: Expressed confusion regarding Figure 1.
Reviewer JmX3/GZCq: Requested a more detailed discussion of related works.
Reviewer GZCq: Advised on writing improvement, clarifying contributions of GRAD, and explaining the motivation behind temporally-coupled perturbations.

## Paper Updates
(All updates in our paper are highlighted in purple.)
* **Introduction, problem formulation and method writing.**
We followed the suggestion from Reviewer mjop and GZCq to update the writing in introduction, preliminaries and methodology sections. Specifically, we added a problem formulation section to provide clear problem settings and revised the illustration of our figure and method.
* **Related works.**
In Appendix B, we included related works mentioned by Reviewer JmX3 and provided detailed discussions on the differences between our work and prior works that also use zero-sum game formulation.
* **Additional experiments.**
We added additional experiments in Appendix C, as recommended by Reviewer mjop and JmX3. These experiments include an extended setting for temporally-coupled perturbations and an evaluation of our approach under transition uncertainty.

## Summary of Novelty and Core Contributions
In response to concerns about the novelty and contributions of GRAD compared to other robust RL works, we would like to reiterate key points:

* **Temporally-Coupled Adversarial Attacks**: We introduce a novel temporally-coupled threat model. These attacks offer insight into the realistic vulnerabilities overlooked by previous threat models.
* **Game-Theoretic Robust Training: GRAD**: We highlight the potential advantages of GRAD in terms of convergence and policy exploitability, which is adaptable to adversaries with different constraints. GRAD is a general and flexible solution for adversarial RL on diverse attack domains.
* **Empirical Efficacy**: Our contributions are substantiated by extensive empirical evaluations showcasing the efficacy of our approach against both temporally-coupled attacks and standard attacks in diverse attack domains.

We sincerely believe that these contributions collectively enhance the field's understanding of adversarial robust RL and introduce a realistic adversarial setting with temporally-coupled perturbations. The broad applicability of GRAD and comprehensive evaluation demonstrate its significant value.

We greatly appreciate all reviewers’ suggestions and hope that our paper updates and responses have effectively addressed their questions and concerns. We eagerly await further discussion and feedback from the reviewers.

---

### Author Response · Authors · 2023-11-19
**Any additional questions or concerns?**

We appreciate the constructive and insightful feedback from all reviewers. In response, we have conducted additional experiments and provided clarifications to comprehensively address the raised concerns, recognizing their significance in enhancing the quality of our paper. We are eager to engage in further discussions with the reviewers during the rebuttal period. If you have any additional questions or concerns, we are more than happy to address them. Looking forward to your response!

---

### Author Response · Authors · 2023-11-23
**Summary of the author-reviewer discussion**

Dear all reviewers and Area Chair,

We would like to express our gratitude for the prompt responses and the thoughtful reconsideration of scores by all the reviewers.

**Reviewer mjop**'s insightful questions during the rebuttal phase help us further clarify **why GRAD stands out as a more general and flexible solution for robust RL, owing to its efficacy and adaptability**. We value the reviewer's patience and ongoing engagement in constructive discussions and are thankful for their consideration in raising the score in the reviewer-AC discussion.

**Reviewer JmX3** acknowledged and appreciated our responses, expressing a willingness to update the score after discussions. We are grateful to the reviewer for **emphasizing the significance of our contribution to the robust RL community**. We sincerely appreciate Reviewer JmX3 for their acknowledgment and valuable reviews.

In response to **Reviewer GZCq**, we **highlighted the theoretical and empirical excellence of our method compared to other robust RL approaches**. Encouragingly, the reviewer increased the score following our revision and clarification. We remain open to addressing any remaining concerns the reviewer may have. Thanks to **Reviewer GZCq** for the detailed feedback and discussion.

During the rebuttal period with reviewers, we also updated the related work section in the Appendix according to the discussion with **Reviewer mjop**.

We believe that we have addressed all concerns raised by the reviewers. We appreciate the valuable insights, active participation in discussions, and positive encouragement from reviewers.

Best regards,

Paper5179 Authors

---

### Meta-Review · Area_Chair_mVZZ · 2023-12-08

**Metareview:**

The paper studies robust RL and proposes GRAD which frames a partially observable two-player zero-sum game and uses Policy Space Response Oracles (PSRO) to achieve robustness against evolving adversarial perturbations. During the discussion, the authors clarified their core contributions (although robust RL has been viewed as a two-player game before, GRAD attempts to find an approximate equilibrium against adversarial policy sets rather than alternating optimization nor best response dynamics). The reviewers requested better motivation of the temporally coupled adversary threat model, that the authors clarified. All the reviewers agreed that the paper was substantially improved after the additions during the rebuttal phase.

**Justification For Why Not Higher Score:**

Reviewers pointed out the inadequate discussion of related work on distributionally robust optimization for robust RL (e.g. uncertainty-aware RL methods). The authors acknowledged their narrower focus on adversarial methods. Reviewers also questioned the novelty or significance of the theoretical convergence results of GRAD, since it is a reiteration of PSRO's known theoretical properties. The authors emphasized that the application of PSRO for robust RL was novel, even if the theoretical ideas underpinning the convergence guarantees were not.

**Justification For Why Not Lower Score:**

The authors substantially improved the paper during the rebuttal phase, including additional experiments and meticulously addressing each of the concerns raised by the reviewers. All of the reviewers increased their score after discussion, with a consensus emerging for marginally above the bar for publication.

---

### Decision · Program_Chairs · 2024-01-16

Accept (poster)